



# Mapping 10-m global impervious surface area (GISA-10m) using multi-source geospatial data

Xin Huang[1,2], Jie Yang[1], Wenrui Wang[1], Zhengrong Liu[1]

[1] School of Remote Sensing and Information Engineering, Wuhan University, Wuhan, 430079, P.R. China
[2] State Key Laboratory of Information Engineering in Surveying, Mapping and Remote Sensing,
Wuhan University, Wuhan, 430079, P.R. China

*Correspondence to*: Jie Yang (yang9tn@163.com)

**Abstract.** Artificial impervious surface area (ISA) documents human footprints. Accurate, timely, and detailed ISA datasets are therefore essential for global climate change and urban planning. However, due to the lack of sufficient training samples
and operational mapping methods, global ISA mapping at 10-m resolution is still lacking. To this end, we proposed a global ISA mapping method leveraging multi-source geospatial data. Based on the existing satellite-derived ISA maps and the crowdsourcing OpenStreetMap (OSM), 58 million training samples were extracted via a series of temporal, spatial, spectral, and geometric rules. Combined with over 2.7 million Sentinel optical and radar images on the Google Earth Engine, we produced the 10 m global ISA dataset (GISA-10m). Based on the test samples that are independent to the training set, GISA-
10m embraced an overall accuracy greater than 86%. In addition, the GISA-10m was comprehensively compared with the existing global ISA datasets, and the superiority of GISA-10m was demonstrated. It was found that China and the United States embraced the largest ISA and road area. The global rural ISA was 2.2 times that of urban while rural road area was 1.5 times larger than that of urban region. The global road area accounted for 14.2% of the global ISA, 57.9% of which was located in the top ten countries. Generally, the produced GISA-10m dataset and the proposed sampling and mapping method
are able to achieve rapid and efficient global mapping, and have potential for detecting other land covers. It was also indicated that global ISA mapping can be improved by incorporating refined OSM data. GISA-10m can be used as a fundamental parameter for Earth system science, and provide valuable support for of urban planning and water cycle study. The GSIA-10m can be freely downloaded from http://doi.org/10.5281/zenodo.5791855 (Huang et al, 2021).

## 1 Introduction

The land dominated by humans has expanded rapidly over the past decades (Friedl et al., 2010; Goldewijk, 2001), resulting in a large amount of terrestrial surface covered by impervious surface area (ISA) (Gong et al., 2020a). ISA is mainly composed of artificial materials, such as gravel, glass, asphalt, and metals (Tian et al., 2018). ISA prevents or decelerates water infiltration while blocks evapotranspiration, which affects the terrestrial water cycle and thermal environment (Qin et al., 2018; Yang et al., 2019). With more attention attracted to the impact of urban sprawl on the global climate environment
(United Nations, 2016, 2018), the global monitoring of ISA would depict the anthropic implications on the water cycle, land



cover, biodiversity (Leng et al., 2015; Li et al., 2020a; Qin et al., 2017). In addition, ISA morphology is also an important parameter for urban planning, socio-economics and population studies (Voss, 2007). In summary, accurate and timely monitoring of global ISA dynamics is valuable for urban habitability (Herold et al., 2006), sustainable development (Dewan and Yamaguchi, 2009), and terrestrial ecosystem services (Goetz et al., 2003).

The global ISA monitoring via satellite remote sensing data has long been recognized. Early efforts usually focused on global ISA mapping using coarse-resolution data, e.g., DMSP (Defense Meteorological Satellite Program) and MODIS (Moderate Resolution Imaging Spectroradiometer) data (Friedl et al., 2010; Huang et al., 2021b; Small et al., 2005). With the free availability of Landsat data and advances in geospatial cloud platforms (e.g. Google Earth Engine, GEE), recent studies have focused on global annual ISA mapping at 30 m (Gong et al., 2020b; Gorelick et al., 2017; Liu et al., 2020b; Woodcock

et al., 2008). For instance, Huang et al., (2021b) generated the global annual ISA dataset GISA (Global Impervious Surface Area) from 1972 to 2019 using over three million Landsat data. Although efforts have been paid to the global ISA monitoring, few studies focused on global ISA mapping at 10-m resolution. Recently, Corbane et al., (2021) generated the Global Human Settlement Layer (GHSL2018) using Sentinel-2 composites and convolutional neural networks. However, GHSL2018 focused more on human settlements and lacks depiction of ISA such as transportation facilities. In addition to

these thematic datasets, ISA was also documented in land cover products. For example, Gong et al., (2019) obtained a land cover map FROM_GLC10 (10-m Finer Resolution Observation and Monitoring of Global Land Cover) for 2017 using Sentinel-2 images. However, the accuracy of ISA in the land cover datasets may not sufficient to meet the needs of global climate change studies and urban planning (Gong et al., 2020b). Therefore, the 10 m global ISA thematic datasets are in urgent need to support various fine-scale applications.

Synthetic Aperture Radar (SAR) performs well in the case of ISA mapping due to its clear response to high-rise buildings and ability to penetrate clouds (e.g. Sentinel 1) (Zhang et al., 2014). SAR data is potential for reducing the common false alarms derived from the optical images, such as bare soil, but it can be affected by complex terrain and shadows. Therefore, existing literatures have invested the collaboration of radar and optical data to improve ISA mapping. For example, Zhang et al., (2020) combined Landsat-8 and Sentinel-1 data to produce a 30 m global ISA dataset (Global Land Cover Fine

Classification System, GLCFCS). Similarly, Marconcini et al., (2020) used Landsat-8 and Sentinel-1 data to outline the world settlement footprint (World Settlement Footprint, WSF) based on support vector machine classifier. Although the current studies have demonstrated the effectiveness of combining multi-source (e.g. radar and optical) remote sensing data for ISA mapping, they usually focus on regional or national scales (Ji et al., 2020; Lin et al., 2020). In addition, combining data with different resolutions for ISA mapping may increase the uncertainty of results. In particular, both Zhang et al.,

(2020) and Marconcini et al., (2020) generated global ISA (or settlement) datasets by using Landsat-8 and Sentinel-1 data, but their resolutions were different, 30 m and 10 m, respectively (Table 1). Generally, 10-m global ISA mapping based on the multi-source remote sensing data (e.g. Sentinel-1 and 2) have been insufficiently investigated in the current literature (Table 1).



**Table 1. The existing global ISA datasets.**

| Name and abbreviation | Data and time span | Nominal resolution | Source of training sample | Classification method and strategy | Type Definition |
|---|---|---|---|---|---|
| Global Impervious Surface Area 30 m, *GISA* (Huang et al., 2021a) | Landsat; 1972—2019 | 30 m | MODIS land cover, Climate Change Initiative land cover, GHSL, FROM_GLC | Random forest classifiers via hexagonal partition | Artificial impervious surface |
| Global Artificial Impervious Area, *GAIA* (Gong et al., 2020b) | Landsat; 1985—2018 | 30 m | Visual interpretation | An exclusion-Inclusion approach via 3.5° grid | Artificial impervious area |
| Global Annual Urban Dynamics, *GAUD* (Liu et al., 2020b) | Landsat; 1985—2015 | 30 m | GAIA, GHSL, Global Urban Land, Global Urban Footprint | Random forest classifiers via 1° grid; temporal segmentation | Urban extent |
| Global Human Settlement Layer, *GHSL2018* (Corbane et al., 2021) | Sentinel-2; 2018 | 10 m | Microsoft building footprint, Facebook settlement, European settlement map, GHSL | Convolutional neural network models within Universal Transverse Mercator zones | Human settlement |
| Finer Resolution Observation and Monitoring of Global Land Cover, *FROM_GLC 10* (Gong et al., 2019) | Sentinel-2; 2017 | 10 m | Visual interpretation | Random forest classifiers | Impervious surface |
| World Settlement Footprint, *WSF2015* (Marconcini et al., 2020) | Landsat-8, Sentinel-1; 2015 | 10 m | Thresholding for spectral index, radar and slope data | SVM classifiers via 1° grid | Human settlement |
| Global Land Cover with Fine Classification System, *GLCFCS* (Zhang et al., 2020a) | Landsat-8, Sentinel-1; 2015 | 30 m | GlobeLand30 | Random forest classifiers via 5° grid | Impervious surface |
| Global Impervious Surface Area 10 m, *GISA-10m* (this study) | Sentinel-1, Sentinel-2; 2016 | 10 m | GISA, OSM GlobeLand30, FROM_GLC10 | Random forest classifiers via hexagonal partition | Artificial impervious surface |

From the perspective of global ISA mapping method, supervised classification has been widely employed (Table 1). The quality of training samples is the major factor affecting the classification results (Foody, 2009). Visual interpretation and automatic extraction from existing datasets are two common methods to generate training samples. Visually-interpreted samples are usually accurate but labour-intensive. Therefore, it is often used for classifications at regional scale (Yang et al.,

2020). On the other hand, samples generated from existing datasets have been proved to be efficient for global ISA mapping in recent years (Marconcini et al., 2020; Zhang et al., 2020a). In fact, ISA samples are diverse, as their response to different sensors varies with materials, geometry, atmospheric conditions, and viewing angles. Therefore, accurate and sufficient samples are required to address the above issue for the purpose of consistent ISA mapping at the global scale. Given the higher spatial resolution (10 m) of the Sentinel satellites, it remains challenging to obtain high-quality and adequate training

samples for 10-m global ISA mapping.

In general, due to the difficulty of collecting training samples and the limitation of computational and storage capacity to deal with the massive data, efficient methods and accurate datasets regarding 10 m global ISA mapping are lacking. Therefore, in this study, we proposed a global ISA mapping method that leverages multi-source geospatial data to mapping 10-m global impervious surface area (GISA-10m). To our knowledge, this was the first global 10 m ISA mapping based on





Sentienl-1 and 2 data. Specifically, combining the multi-source remote sensing and the crowdsourcing OpenStreetMap data, we proposed a sample generation method involving a series of temporal, spatial, spectral, and geometric rules to collect training samples with global coverage. Besides, an adaptive hexagonal partition strategy was used for multi-source feature extraction and classification. Finally, the accuracy of GISA-10m was assessed by three independent sample sets. Meanwhile, we compared the GISA-10m with existing datasets to better reflect its quality, and the ISA distribution in the global urban and rural regions was analysed. Ablation experiments were further conducted to demonstrate the feasibility of OSM data in global ISA mapping.

## 2 Data

### 2.1 Remote sensing data

Sentinel-2 optical and Sentinel-1 SAR data were used in the GISA-10m mapping. Sentinel-2 is a high-resolution multispectral imaging mission by European Space Agency (ESA) Copernicus program. The first Sentinel-2 satellite (Sentinel-2A) has been acquiring high-resolution Earth observation data since June 2015, consisting mainly of four 10-m resolution visible and near-infrared bands, six 20-m resolution red-edge and shortwave infrared bands, and three 60-m bands (Drusch et al., 2012; Zhang et al., 2018). After tested and adjusted, complete global coverage was obtained for Sentinel-2 satellite in 2016 (Fig. S2). Therefore, we used all available Level-1C top of atmosphere (TOA) reflectance data acquired in 2016 for our 10-m ISA mapping. The systematic radiometric calibration, geometric and terrain correction have been performed for the Level-1C TOA data by the ESA. Clouds and shadows were removed via the quality band to obtain cloud-free pixels.

Sentinel-1A was launched on April 2014, carrying a C-band synthetic aperture radar. After the launch of Sentinel-1B in 2016, two satellites had a return visit period of six days at the equator. We used all available Ground Range Detected (GRD) images acquired under Interferometric Wide (IW) mode, with a spatial resolution of 10 m. The boundary noise removal, thermal noise removal, radiometric calibration and terrain correction has been conducted by the GEE with the same processing as Sentinel-1 Toolbox. Sentinel-1 data in both ascending and descending orbit were considered. For the places where two orbits were available, only the descending data was used to avoid the terrain distortion caused by the combination of two orbits. In total, over 2.7 million Sentinel images were used to cover the global terrestrial surface (Fig. S2).

### 2.2 Volunteered geographic information

Volunteered geographic information (VGI) is the geographic information that was created, edited and updated by volunteers (Goodchild, 2007). The well-known VGI project, OpenStreetMap (OSM), provides online maps that can be edited and used by everyone. Since its launch in 2004, OSM has been updated and maintained by seven million volunteers (Haklay and Weber, 2008). OSM has been used for positioning and navigation, urban modeling, and land cover mapping (Fonte et al., 2020; Goetz, 2013; Tian et al., 2019). In fact, over 600 million buildings and roads were tagged in the OSM data





(https://taginfo.openstreetmap.org/keys, last access: 17 Aug 2021). These data should be important reference for ISA mapping, but unfortunately, in the current literature, they have seldom been used for ISA mapping at the global scale. Therefore, we used OSM data as a source of the training samples for the GISA-10m mapping. Specifically, we extracted the buildings and road networks as potential training samples from the OSM planet data built on January 2, 2017[1].

**2.3 Existing ISA datasets**

We intercompared GISA-10m with existing datasets, including GISA, GAIA, GAUD, WSF2015, FROM_GLC10, GLCFCS, GHSL2018 (Table 1). GISA, GAIA and GAUD are Landsat-derived annual global ISA datasets for the time periods 1972-2019, 1985-2018 and 1985-2015, respectively (Gong et al., 2020b; Huang et al., 2021a; Liu et al., 2020b). GHSL2018 is a global settlement layer based on Sentinel-2 composite, where convolutional neural network was used to estimate the 120 settlement probability (Corbane et al., 2021).WSF2015 and GLCFCS are global ISA datasets based on Landsat-8 and Sentienl-1 data. WSF2015 collected samples based on a set of spectral and topographic rules, and GLCFCS derived samples from GlobeLand30 (Marconcini et al., 2020; Zhang et al., 2020a). Gong et al., (2019) generated the 10-m global land cover product FROM_GLC10 using Sentinel-2 data and the random forest classifier. It should be noted that these datasets were different for the definitions of land cover categories and mapping purposes. For instance, GHSL2018 and WSF2015 focused 125 on human settlements, while GAUD delineated urban extent (Table 1). In this study, GISA-10m monitored impervious surface area (ISA) generated by human activities, including all kinds of human settlements, transportation facilities, industries and mining places, by courtesy of the employment of high spatial resolution satellite data. Therefore, artificial impervious surface and human settlements were treated as ISA in this paper.

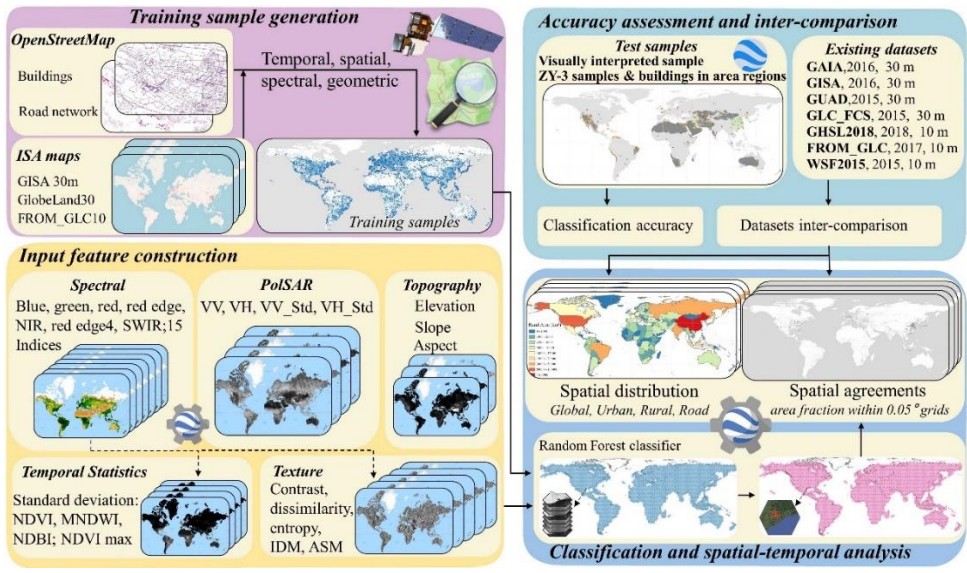

**Figure 1. The flowchart for GISA-10m mapping.**

---

[1] https://planet.openstreetmap.org/planet/2017/planet-170102.osm.bz2, last access: 13 Mar 2021



## 3 Methodology

The main objectives of this study were to: 1) investigate the 10m global ISA mapping (GISA-10m) by combining Sentinel-1 and -2 images with other geographic information; and 2) analyse the distribution of urban and rural ISA at 10-m resolution. The flowchart for GISA-10m mapping was shown in Fig. 1, including training sample generation, multi-source feature
construction, random forest classification, accuracy validation and dataset comparison. Based on satellite-derived ISA maps and VGI (i.e. OpenStreetMap), we proposed a rule-based approach to automatically generate global training samples. Using more than 2.7 million Sentinel images on the GEE, multi-source features were then constructed and fed to random forest classifier to obtain the result. The accuracy of GISA-10m was assessed by visually-interpreted and the third-party samples. To better evaluate the performance of GISA-10m, we compared it with the current state-of-the-art global ISA datasets (Table
1). Finally, the distribution of ISA over urban and rural regions was analysed.

### 3.1 Global ISA mapping using multi-source geospatial data

#### 3.1.1 Sample collection

In the case of large-scale supervised classification, both the quantity and quality of samples are important (Foody and Arora, 1997). ISA is a highly variable object, and its attributes in the Sentinel-2 multispectral images are related to materials,
viewing angles, and atmospheric conditions, while its response to the Sentinel-1 SAR depends on dielectric properties, geometry, and surface roughness. Hence, a large number of training samples were required to address the aforementioned challenges that would be encountered at the global scale. Training samples were usually acquired by means of visual interpretation or automatic extraction from existing datasets. The visual interpretation methods were labour and time intensive, even for small regions. Therefore, at a large scale, training samples were usually extracted from existing datasets
with similar temporal and spatial coverage. However, the sample quality was affected by the quality of the datasets used (Jokar Arsanjani et al., 2016; Wessels et al., 2016). Theoretically, samples extracted from single dataset may result in more errors and uncertainties, while multiple sources can improve the reliability of training samples (Huang and Zhang, 2013). We thus proposed to collect global training samples by incorporating existing ISA datasets and crowdsourcing OSM. To concisely distinguish the two types of ISA samples, we named the ISA sample extracted from the existing satellite-derived
ISA dataset as $ISA_{RS}$ and those extracted from the OSM as $ISA_{OSM}$.

The existing ISA datasets generally covered broad terrestrial surface, but they were different in term of definitions, spatial resolutions, and temporal coverage. In this study, GISA, FROM_GLC10, and GlobeLand30 were chosen to extract training samples due to the following reasons: 1) GISA aimed at mapping global impervious surface area, which was consistent with GISA-10m; 2) GlobeLand30 employed extensive visual interpretation to detect artificial surfaces, which can effectively
reduce false alarms from other datasets (Chen et al., 2015); 3) The definition of FROM_GLC10 (impervious surface) was also consistent with GISA-10m and its spatial resolution was 10 m. GHSL2018, WSF2015 and GAUD were not considered



since they aimed to outline human settlements or urban extents (Table 1). We then collected the eligible training samples according to the following rules.

(1) Temporal rule: GISA was a global ISA dataset during 1972–2019, and we selected its result of 2016 to match the time when Sentinel data was used in this research. GlobeLand30 documented global land cover map for 2000, 2010 and 2020, and here, the 2010 map was chosen. Although the 2020 map was more recent to 2016, it contained ISA that was built after 2016, making it unsuitable for GISA-10m mapping. The following spatial and spectral rules were used to remove the possible errors.

(2) Spatial rule: We first checked the class labels of the three datasets at each pixel. If these labels were the same (i.e. ISA), the pixel was taken as a potential $ISA_{RS}$ sample. The collaboration of multiple datasets can effectively reduce the errors that existed in a single dataset. In addition, we filtered out the edge pixels in each dataset to reduce the uncertainty, since they were more likely to be mixed pixels.

(3) Spectral rule: After the above steps, there may still be a small amount of errors in the current samples. Hence, we applied the spectral rule to remove these erroneous samples. Specifically, we measured the mahalanobis distance between each $ISA_{RS}$ sample to the spectral average of each hexagon (the mapping unit adopted in this study), and filtered out the samples with distance greater than $\mu + \delta$ ($\mu$ and $\delta$ represents the mean and standard deviation, respectively) (Huang et al., 2021a). Vegetation and water bodies were common false alarms in existing datasets (Figs. 2a&b). However, these errors often accounted for a relatively small proportion, and they can be effectively identified and reduced by the spectral rule. It can be seen that most of the water bodies and vegetation (e.g. red rectangles in Fig. 2) were successfully removed from the initial $ISA_{RS}$ training samples.

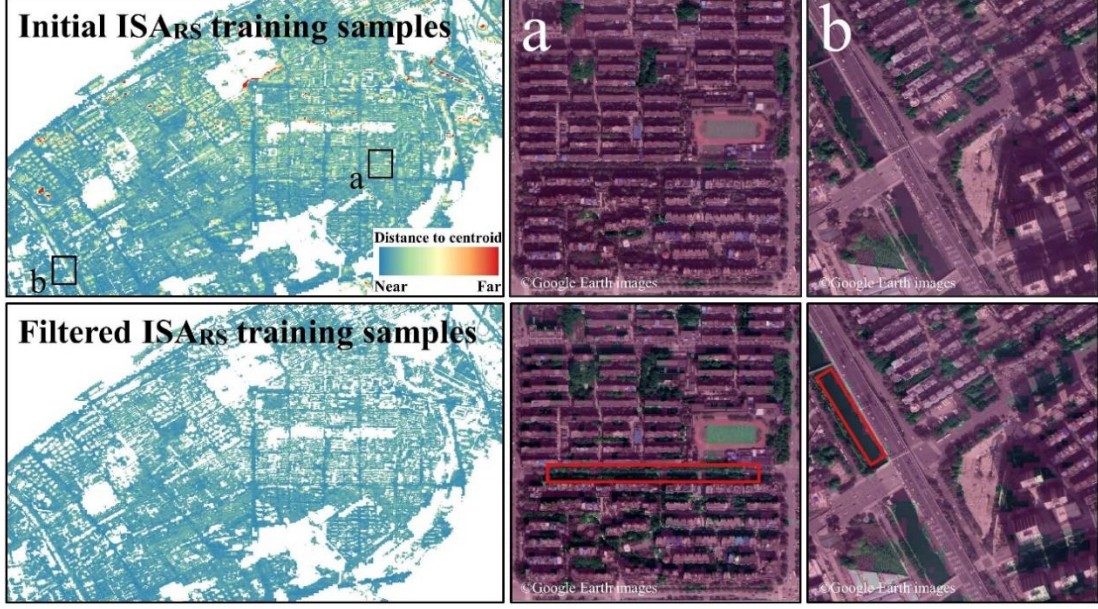

**Figure 2. The example of initial and filtered $ISA_{RS}$ training samples in Wuhan city (30.625382° N, 114.392682° E). The purple in the close-up maps represents the samples.**





On the other hand, we extracted ISA$_{OSM}$ samples from OSM buildings and roads through the following rules.

(1) Temporal rule: We chose the OSM data built on 2 January, 2017 in terms of the time of GISA-10m. This version of OSM data was employed to ensure that buildings and roads were constructed in 2016 or before, and hence, it was suitable for 2016 ISA mapping.

(2) Geometric rule: A natural way to extract training points from OSM was to generate random points within the building or road polygons (Johnson and Iizuka, 2016; Liu et al., 2020a). However, random points may contain erroneous or mixed pixels.

Such problems can be mitigated by making negative buffers to the polygons (Liu et al., 2020a). However, this approach was very time-consuming when applied to global ISA mapping, especially given more than 200 million buildings in the OSM data. Therefore, in this study, we extracted the geometric center of a building polygon as an ISA$_{OSM}$ sample, which was more efficient than buffering and random points. Notably, although we can filter out the erroneous buildings using attribute tags (e.g. dams, swimming pools, playgrounds), the geometric center of a building was not always an ISA sample. Hence, we

further required that the geometric center must be contained by the building. As in Figs. 3a&b, the incorrect building geometric centers (e.g., the vegetation and water, indicated by the yellow points) were successfully identified and removed by the geometric rule. In addition, we excluded buildings with area less than 100 m$^2$ (~ a Sentinel pixel).

Compared with the widely used 30-m Landsat data, the high-resolution Sentinel data promotes a better delineation of roads. We thereby also extracted ISA$_{OSM}$ samples from the OSM road networks. The OSM roads usually consisted of centerlines

rather than boundaries. Therefore, we extracted the center point of each road rather than its geometric center, as the road ISA samples. Given that the width of low-grade roads may be smaller than 10 m (a Sentinel pixel), we kept only the main roads (highway="primary").

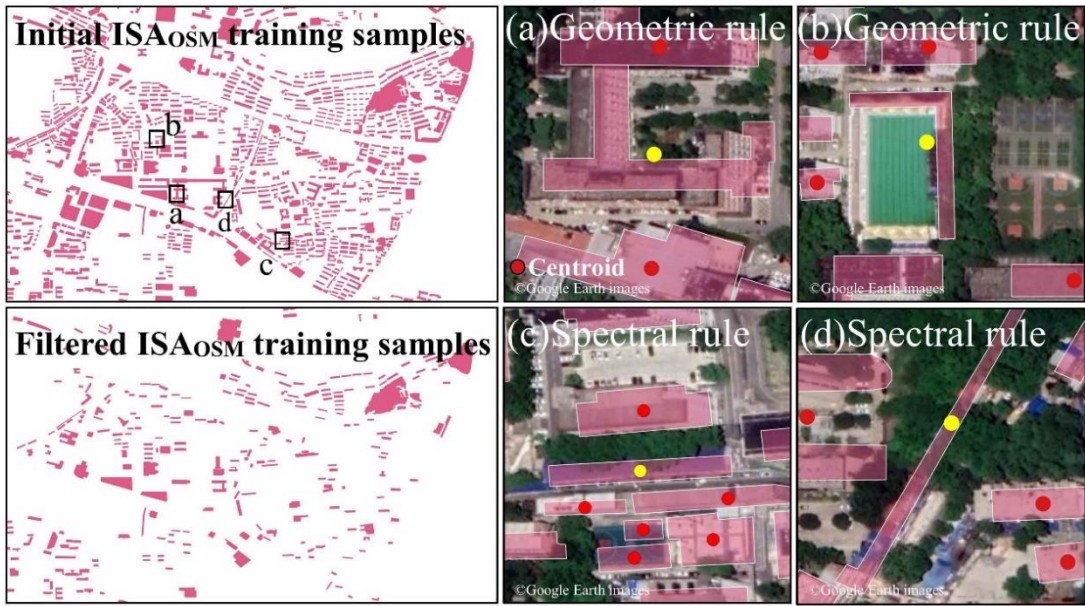

**Figure 3. Examples of initial and filtered ISA$_{OSM}$ training samples in Wuhan city (30.530202° N, 114.356287° E). The yellow points**
**in the close-up maps represent the errors recognized by (a-b) geometric and (c-d) spectral rules.**



(3) Spatial rule: Given the uneven spatial distribution of OSM data (Tian et al., 2019), we then applied the spatial rule to balance its distribution at the global scale. Specifically, for hexagons with more than 10,000 OSM records (i.e. buildings and roads), we randomly selected 10,000 records as initial samples. The dilution of OSM data can significantly reduce subsequent computational cost. In addition, considering that $ISA_{OSM}$ could overlie with $ISA_{RS}$, we removed the $ISA_{OSM}$ samples that were intersected with $ISA_{RS}$.

(4) Spectral rule: Although OSM uses human as sensors, $ISA_{OSM}$ samples may still contain erroneous points, such as vegetation and water body beside roads. As shown in Figs. 3c&d, the yellow points satisfied the temporal, spatial and geometric rules, but they were actually vegetation. Hence, we applied the spectral rule to filter them out. Specifically, the $ISA_{OSM}$ samples whose MNDWI (modified normalized difference water index) or NDVI (normalized difference vegetation index) values larger than $\mu+\delta$ were removed ($\mu$ and $\delta$ represent the mean and standard deviation of the indices, respectively), as these points were more likely to be vegetation or water body (Huang et al., 2021a).

After obtaining the ISA candidate samples, we randomly selected 2,500 $ISA_{RS}$ and $ISA_{OSM}$, respectively, within each hexagon as the final ISA training samples (see Section 5.3 for details). It can be seen that our generated ISA samples covered broad terrestrial surface, especially in India and China where a large number of small villages gather (Fig. 4).

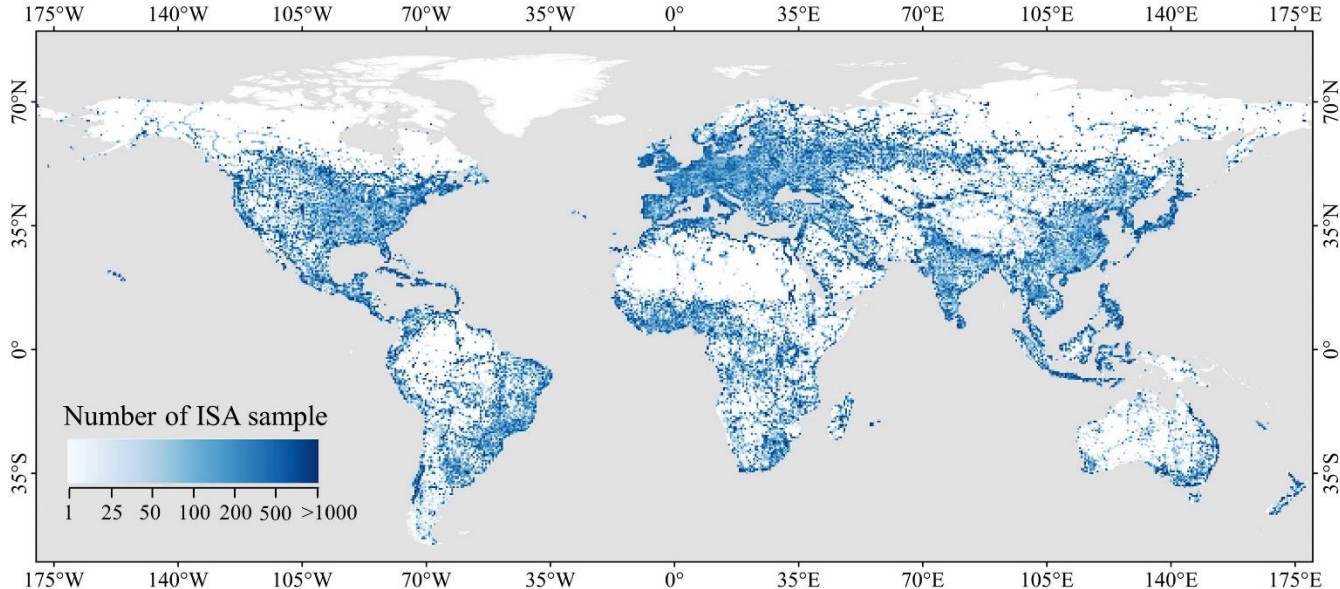

**Figure 4. Global distribution of ISA training samples. The number of samples was counted within 0.5° spatial grid.**

On the other hand, NISA (non-ISA) training samples were also important for accurate ISA mapping. We used the three existing datasets (i.e. GISA 30m, FROM_GLC10, GlobeLand30) and OSM to generate the NISA samples. Firstly, we took the intersect of the NISA regions in the three datasets as the initial NISA sample pool:

$$NISA = NISA_{GISA} \cap NISA_{GlobeLand30} \cap NISA_{FROM\_GLC10} - ISA_{OSM} \tag{1}$$




For GlebeLand30 and FROM_GLC10, NISA was defined as all land cover type other than ISA. We then masked the initial NISA sample pool using OSM buildings and roads to suppress the errors in the existing global dataset. To this end, here we used the OSM version built in December 2020[2], which documented more buildings and road networks than the 2017 version. Besides, we buffered the OSM roads with 30 m to better mitigate the errors. Subsequently, 30,000 points were randomly selected in each hexagon as NISA samples. The distance between each NISA sample was kept larger than 200 m to ensure its diversity and irrelevance. Finally, we extracted 58 million training samples (51,674,533 NISA and 6,897,378 ISA samples) for GISA-10m mapping.

**Table 2. The multi-source features used for GISA-10m mapping.**

| Type | Features | Description | Dimension | Source |
|---|---|---|---|---|
| Spectrum | Blue, green, red, red edge1, red edge2, red edge3, NIR, red edge4, SWIR1 and SWIR2 | 50th percentile value of reflectance derived from all available Sentinel-2 images | 10 | Sentinel-2 |
| Normalized indices | Index1, Index2, Index3, Index4, Index5, Index6, Index7, Index8, Index9, Index10, Index11, Index12, Index13, Index14, Index15 | Normalized indices derived from the spectral bands described above. The indices are calculated as: Index1=NI (NIR, blue), Index2=NI (NIR, green), Index3=NI (NIR, red), Index4=NI (NIR, red edge1), Index5=NI (NIR, red edge2), Index6=NI (NIR, red edge3), Index7=NI (NIR, red edge4), Index8=NI (SWIR1, blue), Index9=NI (SWIR1, green), Index10=NI (SWIR1, red), Index11=NI (SWIR1, NIR), Index12=NI (SWIR2, blue), Index13=NI (SWIR2, green), Index14=NI (SWIR2, red), Index15=NI (SWIR2, NIR), where NI represents the function $(b1- b2) / (b1+ b2)$, b1 and b2 denote two spectral bands | 15 | Sentinel-2 |
| SAR | VV, VH | Temporal mean VV and VH backscatter coefficients of Sentinel-1 images | 2 | Sentinel-1 |
| Temporal statistics | NDVI_Std, MNDWI_Std, NDBI_Std, NDVIMax, VV_Std, VH_Std | Standard deviation of NDVI, MNDWI, NDBI, VV and VH backscatter coefficients; Maximum NDIV of the year | 5 | Sentinel-1& Sentinel-2 |
| Texture | Contrast, dissimilarity, entropy, IDM, ASM | The GLCM texture derived from NIR band of Sentinel-2 data, including entropy, dissimilarity, contrast, angular second moment (ASM) and inverse difference moment (IDM) | 5 | Sentinel-2 |
| Topography | Elevation, slope and aspect | Slope and aspect calculated from the elevation | 3 | SRTM & GMTED |

### 3.1.2 Multi-source feature extraction

The dedicated image pyramid of GEE enabled us to perform pixel-wise feature extraction (Gorelick et al., 2017; Huang et al., 2021a; Yang and Huang, 2021). Therefore, based on all available Sentinel data in 2016, we constructed a set of spectral, phenological, texture, SAR and topographical features with the temporal composite method (Table 2). This approach used all

---

[2] https://planet.openstreetmap.org/planet/2020/planet-201207.osm.bz2, last access: 13 Mar 2021





available data and at the same time allowed us to reduce the feature dimension, preserve the temporal information and minimize the effects from clouds and shadows (Yang and Huang, 2021). Firstly, the 50th percentile TOA reflectance was

calculated for each spectral band (Table 2). Moreover, we calculated 15 spectral indexes to improve the discrimination between ISA and NISA (Table 2). These indices were built according to the following criteria: (1) They were mainly constructed by near-infrared (NIR) and short-wave infrared (SWIR) bands due to their better atmospheric transmission (Huang et al., 2021a; Yang and Huang, 2021); (2) Each index contained at least one 10-m band (i.e. visible and NIR bands) to ensure the spatial resolution of the features. To further exploit the textural information for the ISA mapping, we computed

the gray-level co-occurrence matrix (GLCM) via NIR band (Weng, 2012). Owing to the high redundancy among GLCM measurements (Clausi, 2002; Zhang et al., 2020a), we chose the contrast, dissimilarity, entropy, IDM (inverse difference moment) and ASM (angular second moment) for texture extraction (Conners et al., 1984; Haralick et al., 1973; Rodriguez-Galiano et al., 2012). The window size for GLCM measurements was set to $7 \times 7$ as it was suitable for urban classification with image resolution from 2.5 to 10 m (Puissant et al., 2005). Besides, we averaged the GLCM from different directions (0,

45, 90, and 135) to maintain the rotational invariance (Rodriguez-Galiano et al., 2012).

Given that spectral responses of vegetation and water bodies vary over time, we calculated the maximum NDVI as well as standard deviation of NDVI, MNDWI, and NDBI (normalized difference built-up index) to further enhance the temporal information (Tucker, 1979; Xu, 2006; Zha et al., 2003). These temporal characteristics were useful in identifying NISA with temporal fluctuations. For example, the spectra of fallow cropland and ISA were similar, even SAR data may not well separate them. However, the NDVI of cropland can describe the changes of crops growth, and hence, its standard deviation

can be used to distinguish between ISA and cropland. In addition, to increase the robustness of these temporal features, Sentinel-2 data from adjacent two years were also included.

With regard to the SAR data, the VV (vertical-vertical polarization) and VH (vertical-horizontal polarization) backscatter coefficients from the Sentienl-1 images were selected. Specifically, based on all available Sentienl-1 data, the annual mean

and standard deviation of the VV and VH backscatter coefficients were calculated by the temporal composite method:

$$\sigma_{mean} = \frac{1}{n}\sum_{i=1}^{n} \sigma_i \tag{2}$$

$$\sigma_{std} = \sqrt{\frac{\sum_{i=1}^{n}(\sigma_i - \sigma_{mean})^2}{n}} \tag{3}$$

where $n$ denotes the total number of Sentinel-1 observations within a year and $\sigma_i$ represents the $i$th backscatter coefficient observation in the year. The temporal mean method can reduce speckle noise in the SAR images (Lin et al., 2020), while the

annual standard deviation can reflect temporal information. In addition, DEM-derived (Digital Elevation Model) topographic features were also constructed to reduce false alarms induced by SAR data over mountain areas (Ban et al., 2015; Gamba and Lisini, 2013). Specifically, we used SRTM (Shuttle Radar Topographic Mission) in the areas below 58° latitude and GMTD2010 (Global Multi-resolution Terrain Elevation Data 2010) in the areas above 58° (Huang et al., 2021a). Finally, a total of 41 features were constructed on the 2.7 million Sentinel images (2,613,180 Sentinel-2 and 122,156 Sentinel-1) and

DEM data.



### 3.1.3 Hexagon-based adaptive random forest classification

When dealing with global land cover classification, the global terrestrial surface was usually divided into homogeneous sub-regions according to criteria such as climate, land cover or administrative regions (Goldblatt et al., 2018; Homer et al., 2004; Turner, 1989). For global ISA mapping, regular square grids were commonly used (Table 1), such as 1° and 5° grids (e.g.
WSF2015 and GLCFCS). Herein we divided the terrestrial surface into 2° hexagonal grids (Fig. 1), due to its symmetry and invariance (Birch et al., 2007; Goldblatt et al., 2018; Huang et al., 2021a). Besides, there were no gaps or overlaps between hexagons, and the distance between adjacent hexagon centers was approximately equal (Richards et al., 2000).

Random forest (RF) classifier has been widely used in global ISA mapping, due to its robustness to erroneous samples, flexibility to high-dimensional data and tolerance to noise (Bauer and Kohavi, 1999; Wulder et al., 2018) (Table 1). It
utilizes ensemble learning to obtain predictions by voting on categories through multiple decision trees (Breiman, 2001). Each tree uses a random subset of the input features to increase the generalization ability. In addition, trees are grown from different subsets of training data (i.e. bagging or bootstrap) to increase the diversity (Rodriguez-Galiano et al., 2012). RF has been proved to outperform other classifiers when dealing with large-scale and high-dimension data (Gislason et al., 2006; Goldblatt et al., 2016). The flexibility of RF to handle multi-source data also makes it convenient for us to deal with Sentinel
radar and optical data. Therefore, together with the aforementioned multi-source features and global training samples, RF was used for GISA-10m mapping. As suggested by Yang and Huang, (2021), the number of trees was set to 200.

### 3.2 Accuracy assessment

The test samples of GISA-10m included (1) visually interpreted samples via Google Earth, (2) test samples extracted from the ZiYuan-3 (ZY-3) built-up datasets (Liu et al., 2019), and (3) building samples located in the arid areas.
(1) As suggested by Stehman and Foody (2019), we used cluster sampling to collect the visually-interpreted test samples. The primary sampling unit involved 59 grids with a side length of 1°, which was randomly selected based on population, ecoregion, and urban landscape (red squares in Fig. 5). The secondary sampling unit included the random samples within each grid. Specifically, in each grid, we randomly selected 100 ISA and 100 NISA points to test their accuracy. An equal allocation of ISA and NISA test samples could reduce the bias of the accuracy assessment and hence allow for a more
accurate estimate of user's accuracy (Marconcini et al., 2020; Olofsson et al., 2014; Stehman, 2012; Story and Congalton, 1986; Wehmann and Liu, 2015). By referring to the high-resolution Google Earth images, a pixel (10m × 10m) was labelled as ISA if more than half of its area was cover by ISA, otherwise it was identified as NISA. In such way, a total of 10,800 test samples were obtained.

(2) Liu et al., (2019) proposed a multi-angle built-up index to extract built-up areas from ZY-3 images covering 45 global
cities, with an overall accuracy greater than 90%. The multi-angle ZY-3 images depicted the three-dimensional and vertical structure of buildings, which were more effective and accurate than the planar feature extraction for detecting built-up areas. Given the higher spatial resolution (2 m) and better accuracy of ZY-3 global built-up dataset, we extracted test samples from

it in the year of 2016 (Huang et al., 2021a; Liu et al., 2019). A sample (10m × 10m) was labeled as ISA if more than 50% of its area was classified as ISA in the ZY-3 dataset, while NISA samples were those with no built-up pixels in the area (Huang

et al., 2021a). For each city, the number of samples was proportional to the area of the ZY-3 image, and the ratio of ISA and NISA test samples was consistent with the ratio of the built-up and non-built-up (Huang et al., 2021a). In this way, we obtained 47,216 NISA and 21,152 ISA samples (green dots in Fig. 5) from 24 cities in the ZY-3 built-up dataset.

(3) Considering the difficulty of ISA extraction in the arid regions (Tian et al., 2018), we paid special attention to the accuracy assessment in the arid regions. To this end, we visually interpreted 5,385 building pixels in these regions. A total of

25 photo interpreters were recruited for this task by referring to the Google Earth images. These samples were further checked by three experts. The arid regions were defined according to the "Deserts and Xeric Shrublands" biome in Olson et al., (2001).

Based on the three groups of test samples aforementioned, the accuracy of GISA-10m was assessed by overall accuracy (OA), kappa, producer's accuracy (PA), user's accuracy (UA), and F-Score (the harmonic mean of PA and UA).

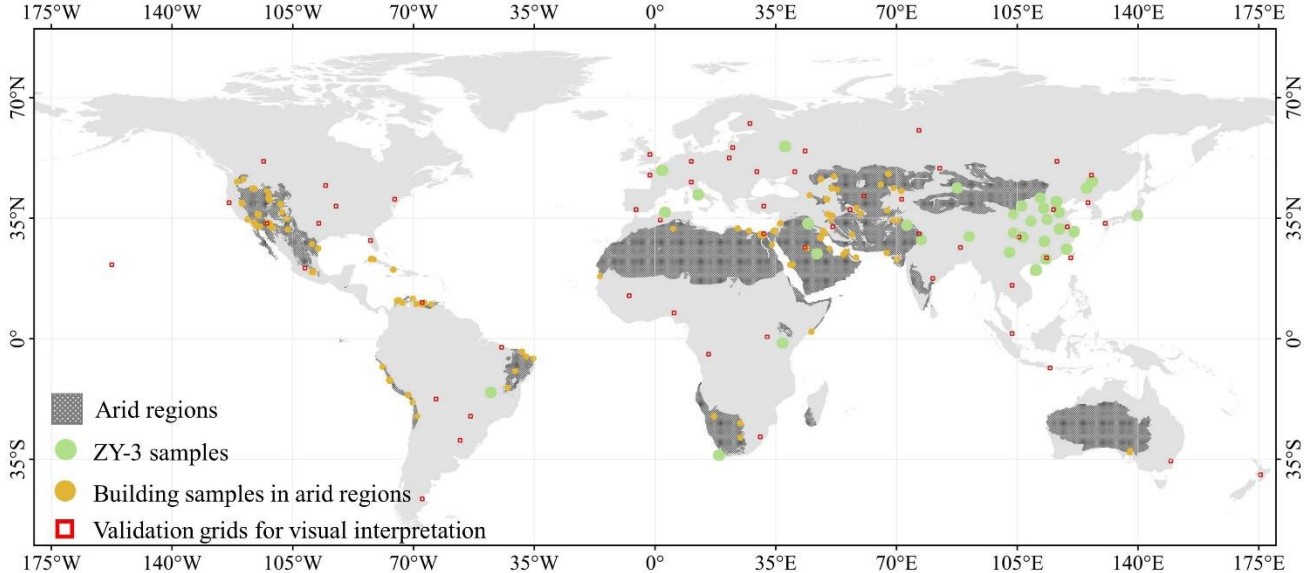


**Figure 5. Global distribution of the test samples and grids used in this study, including (1) 59 grids for visual interpretation, (2) ZY-3 reference set covering 23 cities, and (3) 5,385 building samples in the arid regions. The arid regions were extracted from "Deserts and Xeric Shrublands" biome in Olson et al., (2001).**

### 3.3 Inter-comparison between different global datasets

Seven existing global ISA datasets were used for inter-comparison with GISA-10m, including GHSL2018, GLCFCS, WSF2015, FORM_GLC10, GISA, GAUD, and GAIA (Table 1). First, the three groups of test samples mentioned above were used to assess and compare the accuracy of these products. Second, their spatial agreements with GISA-10m were measured by the linear fit of ISA fraction, including metrics such as correlation coefficient and root mean square error (RMSE). Finally, attention was paid to their performances in urban and rural regions for a comprehensive assessment.

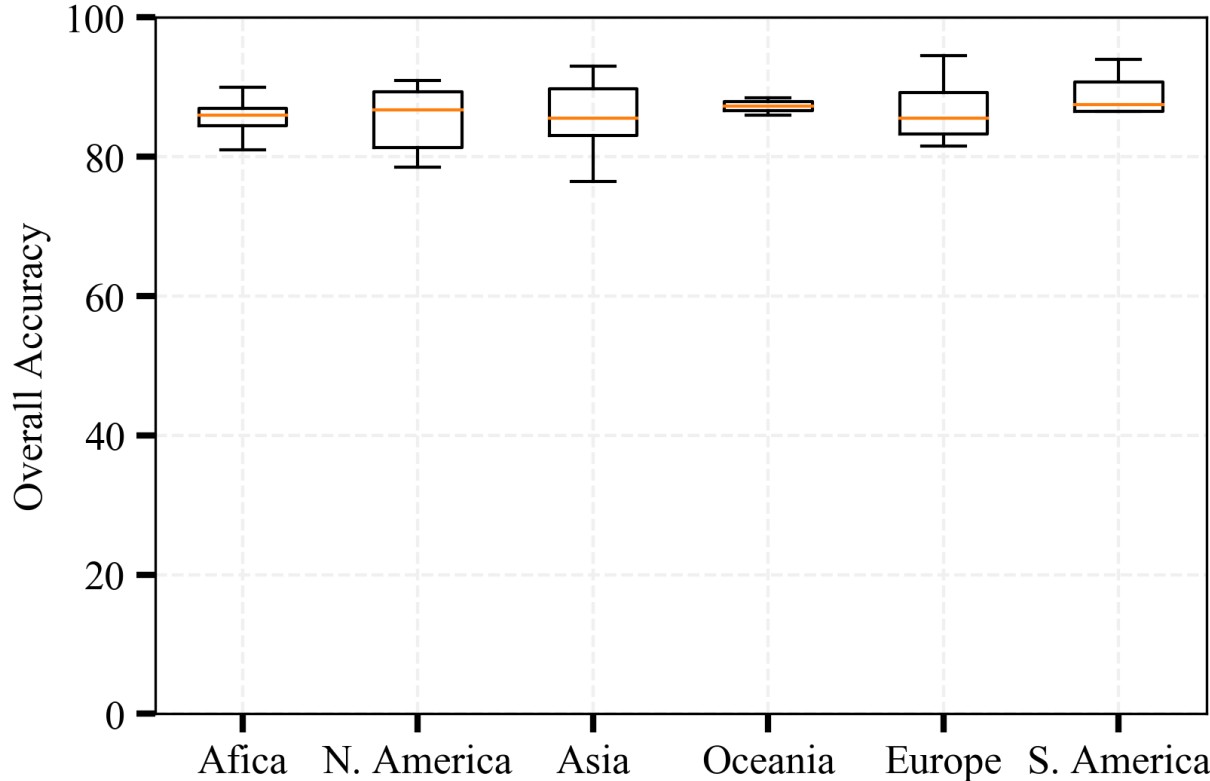

**Figure 6. Box plots of the overall accuracy for GISA-10m in the six continents.**

## 4 Results

### 4.1 Accuracy assessment of GISA-10m

#### 4.1.1 Global scale

The accuracy assessment based on the visually-interpreted samples were shown in the Table 3. GISA-10m exhibited the highest OA of 86.06%, with an increase of +2.73%, +3.73% and +2.3%, respectively, with respect to GHSL2018, GLCFCS and WSF2015 (Table 3). The Kappa of GISA-10m was 0.7165, which exceeded the WSF2015, FROM_GLC10 and GAIA by 0.052, 0.1774 and 0.2039, respectively. Alongside, GISA-10m showed higher accuracy as to the 30 m datasets (i.e. GISA, GAUD, GAIA), which suggested a better delineation of global ISA due to its higher resolution. Fig. 6 summarized the

results of the accuracy assessment at the continent level, with the average and standard deviation of OA for each continent shown in the box plots. In general, GISA-10m exhibited stable performance for each continent, with an average OA more than 85%. Specifically, Oceania and South America obtained the best OA of 87.25% and 87.08%, followed by Europe (86.45%) and Asia (85.85%).

**Table 3. Results of quantitative accuracy assessment via visually-interpreted and ZY-3 samples between GISA-10m and the**
**existing ISA datasets. OA represents the overall accuracy.**

| Globe | Visually interpreted samples (n=10800) | | | | ZY-3 samples (n=68368) | | | |
|---|---|---|---|---|---|---|---|---|
| | OA (%) | Kappa | F-Score of ISA (%) | F-Score of NISA (%) | OA (%) | Kappa | F-Score of ISA (%) | F-Score of NISA (%) |
| GISA-10m | **86.06** | **0.7165** | **83.65** | **88.55** | **86.25** | **0.6664** | 76.25 | **90.32** |
| GHSL2018 | 83.33 | 0.6540 | 78.66 | 86.89 | 84.53 | 0.6401 | 75.27 | 88.74 |
| GLCFCS | 82.33 | 0.6336 | 77.57 | 85.96 | 84.56 | 0.6280 | 73.68 | 89.08 |
| WSF2015 | 83.76 | 0.6645 | 79.68 | 87.06 | 85.44 | 0.6664 | **77.35** | 89.27 |
| FROM_GLC10 | 78.16 | 0.5391 | 69.65 | 83.39 | 83.66 | 0.6082 | 72.39 | 88.39 |
| GISA | 78.84 | 0.5532 | 70.65 | 83.88 | 85.63 | 0.6627 | 76.65 | 89.63 |
| GAUD | 77.36 | 0.5185 | 67.46 | 83.01 | 85.59 | 0.6549 | 75.70 | 89.76 |
| GAIA | 77.05 | 0.5126 | 67.13 | 82.77 | 84.23 | 0.6381 | 75.39 | 88.40 |

GISA-10m obtained the best OA of 86.25% on the ZY-3 samples, outperforming GHSL2018, GLCFCS and WSF2015, by

1.72%, 1.69% and 0.81%, respectively. The ZY-3 images employed by Liu et al., (2019) covered 45 major global cities, and

therefore the ZY-3 samples were more inclined to reflect the accuracy in urban regions. Therefore, the accuracy difference

between various datasets was not significant (Table 3). Due to the relatively coarser resolution, the 30 m datasets usually

tended to overestimate the ISA extent (Gong et al., 2020b), resulting in higher UA but lower PA (Table S1). For example,

the ISA UA of GISA was slightly higher than GISA-10m, but its PA was much smaller than the latter (Table S1). However,

when the two metrics (PA and UA) were considered at the same time (i.e. F-Score), GISA-10m outperformed GISA.

### 4.1.2 Rural and arid regions

The population of rural regions is comparable to that of urban regions (https://data.worldbank.org/). Existing studies as well

as their global ISA datasets usually focus on the performance in urban regions, but the accuracy of rural ISA regions has not

been sufficiently assessed. Hence, in this study, we paid special attention to the accuracy assessment in the global rural

regions. Specifically, we divided the GISA-10m into urban and rural regions using the urban boundary defined by Li et al.,

(2020). In fact, due to the random sampling strategy, most of visually-interpreted test samples were located in rural regions.

In the case of the visually-interpreted samples, GISA-10m exhibited a better OA of 86.19% against the GHSL2018 (84.92%),

GLCFCS (83.25%), FROM_GLC10 (78.83%) and WSF2015 (83.81%). As regards the three 30-m datasets (i.e. GISA,

GAIA, GAUD), their ISA accuracy (F-Score) decreased significantly in the rural regions while the NISA accuracy was

relatively stable (Tables 2&3). Having a closer look at the PA, one can notice that the ISA PA decreases by more than 15%

for all the three 30-m datasets (Table S2), which suggested their more omission errors in the rural regions (Fig.12b). This

demonstrated the deficiency of 30-m datasets in depicting rural ISA and also reflected the importance of 10-m global ISA

mapping.





**Table 4. Results of quantitative accuracy assessment via visually-interpreted and ZY-3 samples in rural regions between GISA-10m and the existing ISA datasets. OA represents the overall accuracy.**

| Rural Regions | Visually interpreted samples (n=9547) | | | | ZY-3 samples (n=43950) | | | |
|---|---|---|---|---|---|---|---|---|
| | OA (%) | Kappa | F-Score of ISA (%) | F-Score of NISA (%) | OA (%) | Kappa | F-Score of ISA (%) | F-Score of NISA (%) |
| GISA-10m | **86.19** | **0.6794** | **77.96** | **90.48** | **90.85** | **0.4768** | 52.46 | **94.94** |
| GHSL2018 | 84.92 | 0.6297 | 73.34 | 89.88 | 88.95 | 0.4656 | **52.82** | 93.74 |
| GLCFCS | 83.25 | 0.5871 | 70.15 | 88.72 | 89.46 | 0.4261 | 48.33 | 94.13 |
| WSF2015 | 83.81 | 0.6012 | 71.17 | 89.12 | 89.37 | 0.4514 | 51.05 | 94.04 |
| FROM_GLC10 | 78.83 | 0.4485 | 57.08 | 86.24 | 88.59 | 0.3884 | 45.08 | 93.63 |
| GISA | 77.87 | 0.4082 | 52.53 | 85.80 | 89.83 | 0.3954 | 44.66 | 94.40 |
| GAUD | 76.38 | 0.3516 | 46.13 | 85.05 | 89.70 | 0.3199 | 36.35 | 94.40 |
| GAIA | 75.41 | 0.3213 | 43.05 | 84.49 | 88.93 | 0.3611 | 41.85 | 93.88 |

**Table 5. Results of quantitative accuracy assessment via visually-interpreted and ZY-3 samples in arid regions between GISA-10m and the existing ISA datasets. OA represents the overall accuracy.**

| Arid Region | Visually interpreted samples (n=1020) | | | | ZY-3 samples (n=10827) | | | |
|---|---|---|---|---|---|---|---|---|
| | OA (%) | Kappa | F-Score of ISA (%) | F-Score of NISA (%) | OA (%) | Kappa | F-Score of ISA (%) | F-Score of NISA (%) |
| GISA-10m | **86.67** | **0.7358** | 86.05 | **88.22** | **89.64** | **0.7296** | **79.95** | **93.01** |
| GHSL2018 | 86.57 | 0.7336 | **86.06** | 87.99 | 85.13 | 0.5817 | 67.68 | 90.34 |
| GLCFCS | 82.16 | 0.6454 | 80.32 | 84.46 | 85.14 | 0.6232 | 72.45 | 89.82 |
| WSF2015 | 82.45 | 0.6516 | 80.95 | 84.56 | 88.37 | 0.6881 | 76.53 | 92.27 |
| FROM_GLC10 | 76.27 | 0.5271 | 70.97 | 80.59 | 84.06 | 0.5755 | 68.18 | 89.37 |
| GISA | 80.20 | 0.6058 | 76.89 | 83.39 | 87.72 | 0.6795 | 76.23 | 91.72 |
| GAUD | 77.06 | 0.5424 | 71.88 | 81.20 | 88.66 | 0.6894 | 76.37 | 92.54 |
| GAIA | 77.45 | 0.5506 | 72.84 | 81.35 | 85.78 | 0.6317 | 72.79 | 90.37 |

Furthermore, we focused on the accuracy assessment in arid regions. In general, the OA of GISA-10m was higher than the existing datasets (Table 5). Although its ISA UA did not always outperform other datasets, GISA-10m achieved the highest PA among the existing ones (Table S3). Specifically, GISA-10m exhibited a notably higher ISA PA compared to GLCFCS,

FROM_GLC10, GISA, GAUD and GAIA (Table S3), indicating its better ability of detecting ISA in arid regions (Fig. 7). Moreover, the accuracy of these global ISA products was assessed using our manually and randomly chosen rural building samples (see Section 3.2). It can be found that GISA-10m extracted detected 15% more buildings in arid regions with respect to FROM-GLC10, GAUD and GAIA (Table S4), which again verified its better performance in describing rural ISA.



Figure 7. Comparison of the GISA-10m and other datasets over arid regions in (a) Kabul, Afghanistan; (b)Mashhad, Iran; (c) Buraidah, Saudi Arabia; (d) Ashkhabad, Turkmenistan. The illustration is of Sentinel-2 images with false-color combination (R: NIR, G: Red, B: Green) to enhance the ISA.



### 4.2 Global ISA distribution

### 4.2.1 Urban and rural ISA

Based on GISA-10m, we analyzed the global ISA distribution at 10-m scale (Fig. 8). Global ISA was mainly distributed in Asia (41.43%), North America (20.59%), Europe (18.93%), followed by Africa (9.78%) and South America (7.50%). It was found that 67% of global ISA was located in the Eastern Hemisphere, while 85% of ISA was distributed in the north of the equator. Rural ISA was more scattered than urban ISA (Fig. 8), and it was mainly located in Asia (42.84%), Europe (19.49%) and North America (16.51%). Asia embraced the largest urban ISA, more than twice as Europe. Although North America

only accounted for 20% of global ISA, its urban ISA took up more than 29% of the global total. Taking a closer look at the ratio of rural and urban ISA (Table 6), one can see that rural ISA were 2.2 times larger than the urban. At the continental level, Africa possessed the highest "rural-to-urban ratio", which may be related to its large population but relatively poor economy.

**Table 6. Impervious surface area derived from GISA-10m in the six continents.**

| ISA | Europe | Africa | S. America | Oceania | N. America | Asia | Globe |
|---|---|---|---|---|---|---|---|
| Total ($10^5$km$^2$) | 1.88 (18.93%) | 0.97 (9.78%) | 0.75 (7.50%) | 0.18 (1.76%) | 2.05 (20.59%) | 4.12 (41.43%) | 9.94 (100%) |
| Rural ($10^5$km$^2$) | 1.33 (19.49%) | 0.78 (11.43%) | 0.55 (8.11%) | 0.11 (1.62%) | 1.13 (16.51%) | 2.93 (42.84%) | 6.84 (100%) |
| Urban ($10^5$km$^2$) | 0.55 (17.69%) | 0.19 (6.16%) | 0.19 (6.17%) | 0.07 (2.07%) | 0.92 (29.56%) | 1.19 (38.35%) | 3.10 (100%) |
| Rural/Urban | 2.42 | 4.08 | 2.89 | 1.73 | 1.22 | 2.46 | 2.20 |



**Figure 8. Spatial distribution of ISA in (a) the global and (b) rural regions. The pixel represents the ISA regions in 0.01° grid while the doted lines denote the cumulative histograms.**

China and the United States (US) embraced 33% of global ISA. Together with Russia, Brazil, India, Japan, Indonesia, France, Canada and Germany, these ten countries accounted for 58% of the world. The urban ISA owned by the top ten countries (US, China, Russia, Brazil, Japan, India, Mexico, France, Germany, and the United Kingdom) took up 69% of the global total, while the top ten countries in terms of rural ISA (China, US, Russia, Brazil, India, Indonesia, Japan, France, Canada, and Germany) accounted for only 54% of the total. In Africa, the Republic of South Africa had much more urban ISA than other countries. However, Nigeria showed comparable rural ISA to the South Africa (~7738 km$^2$). China ranked first in term of rural ISA, most of which was located in the North China Plain (Fig. 9b). Indonesia also possessed much rural ISA, since it ranked sixth in the rural ISA but its urban ISA only ranked sixteenth.

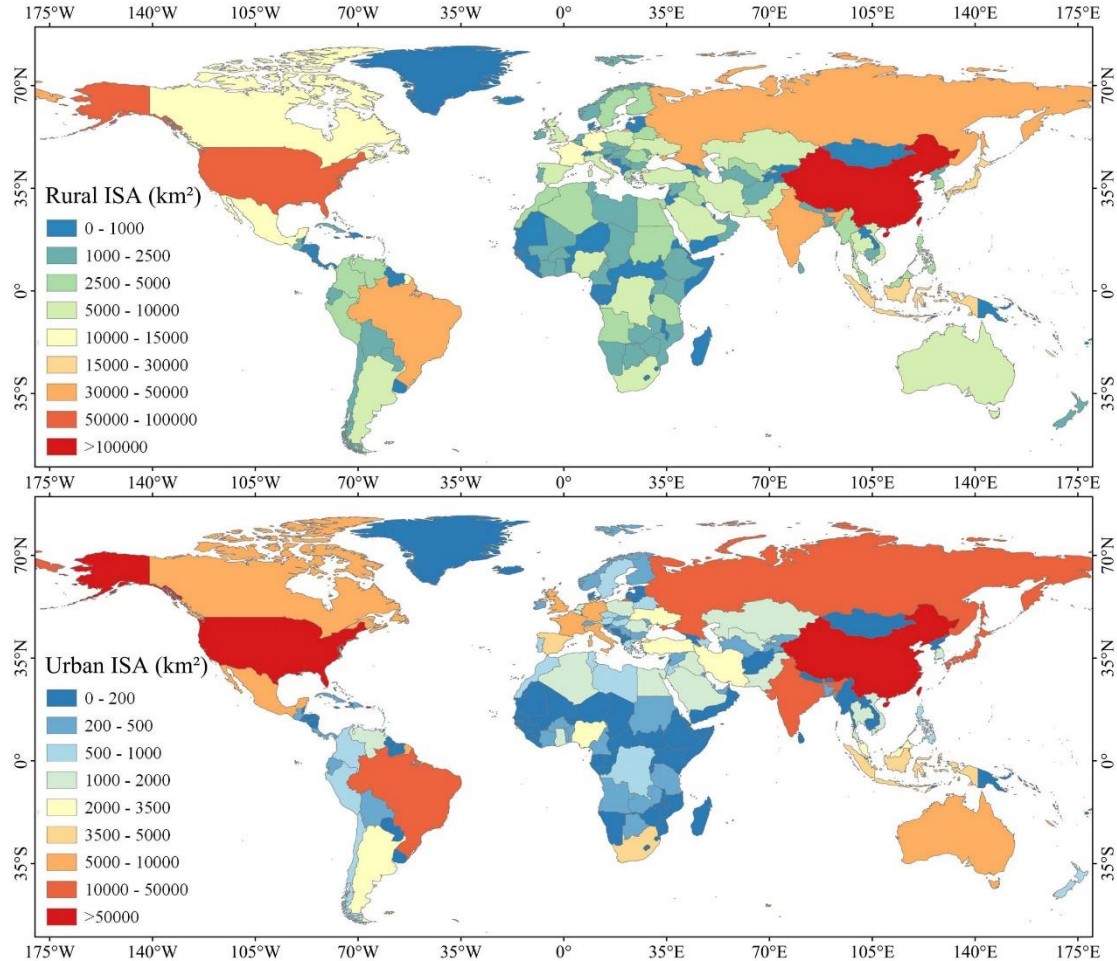

**Figure 9. Urban and rural ISA at the country scale based in GISA-10m.**

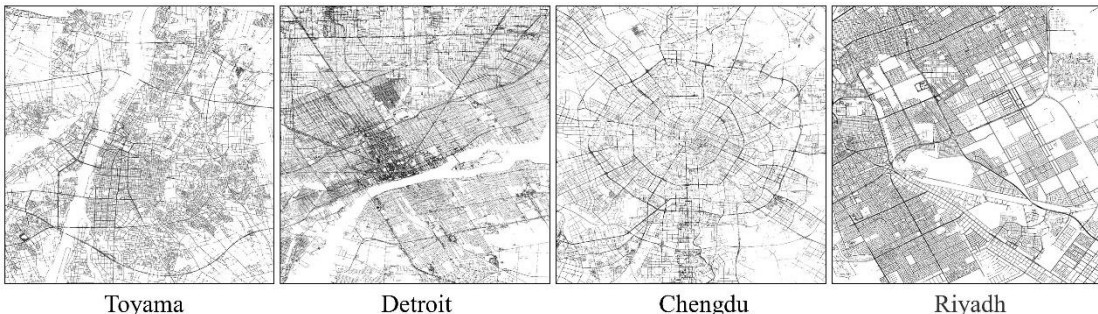

**Figure 10. Examples of road area derived from GISA-10m and OSM in the Toyama (Japan), Detroit (US), Chengdu (China), and Riyadh (Saudi Arabia).**





### 4.2.2 Global road area

Roads are major anthropic footprints, we attempted to analyse the global road area based on GISA-10m, by courtesy of its high spatial resolution. Firstly, the road networks were extracted from OSM, and then, the ISA regions in the GISA 10-m

within a 10-m buffer of the road networks were identified as the road areas (Fig. 10). Results showed that 82.84% of the global road areas located in Asia (30.74%), North America (27.17%) and Europe (24.92%), while the remaining 17.16% was owned by South America (8.26%), Africa (7.47%) and Oceania (1.44%). Although Asia exceeded the other continents regarding ISA and rural road area, it possessed less urban road area than North America. China and US had the largest road area, together accounting for 29% of the global total, which were followed by Brazil, Japan, Russia, Germany, India, France,

Indonesia, and Mexico. The top ten countries owned more than half of global roads. The global road area accounted for 14.18% of the global ISA, and rural road area was 1.5 times larger than urban (Table 7). However, it should be noted that these estimates might be biased owing to the incompleteness of the OSM data. In addition, narrow roads might be partly detected or missed due to the limitation of spatial resolution.

**Table 7. Statistics of road network derived from GISA-10m and OSM in the six continents.**

| Road | Europe | Africa | S. America | Oceania | N. America | Asia | Globe |
|---|---|---|---|---|---|---|---|
| Total ($10^4$km$^2$) | 3.51 (24.92%) | 1.05 (7.47%) | 1.16 (8.26%) | 0.20 (1.44%) | 3.83 (27.17%) | 4.34 (30.74%) | 14.10 (100%) |
| Rural ($10^4$km$^2$) | 2.27 (26.88%) | 0.71 (8.43%) | 0.75 (8.88%) | 0.11 (1.26%) | 1.84 (21.73%) | 2.77 (32.82%) | 8.45 (100%) |
| Urban ($10^4$km$^2$) | 1.24 (21.99%) | 0.34 (6.03%) | 0.42 (7.34%) | 0.10 (1.70%) | 2.00 (35.29%) | 1.56 (27.65%) | 5.66 (100%) |
| Rural/Urban | 1.82 | 2.09 | 1.81 | 1.10 | 0.92 | 1.77 | 1.49 |

**5 Discussions**

### 5.1 Inter-comparison with existing datasets

To further validate the performance of GISA-10m, we compared it with a series of existing state-of-the-art global datasets, including three 10-m datasets (i.e. WSF2015, GHSL2018, FROM_GLC10) and four 30-m datasets (i.e. GLCFCS, GAUD, GAIA, and GISA). The spatial agreements over urban and rural regions were estimated by the linear fitting of ISA fraction.

Considering their difference of spatial resolutions, the ISA fraction was calculated within the 0.05° spatial grid.



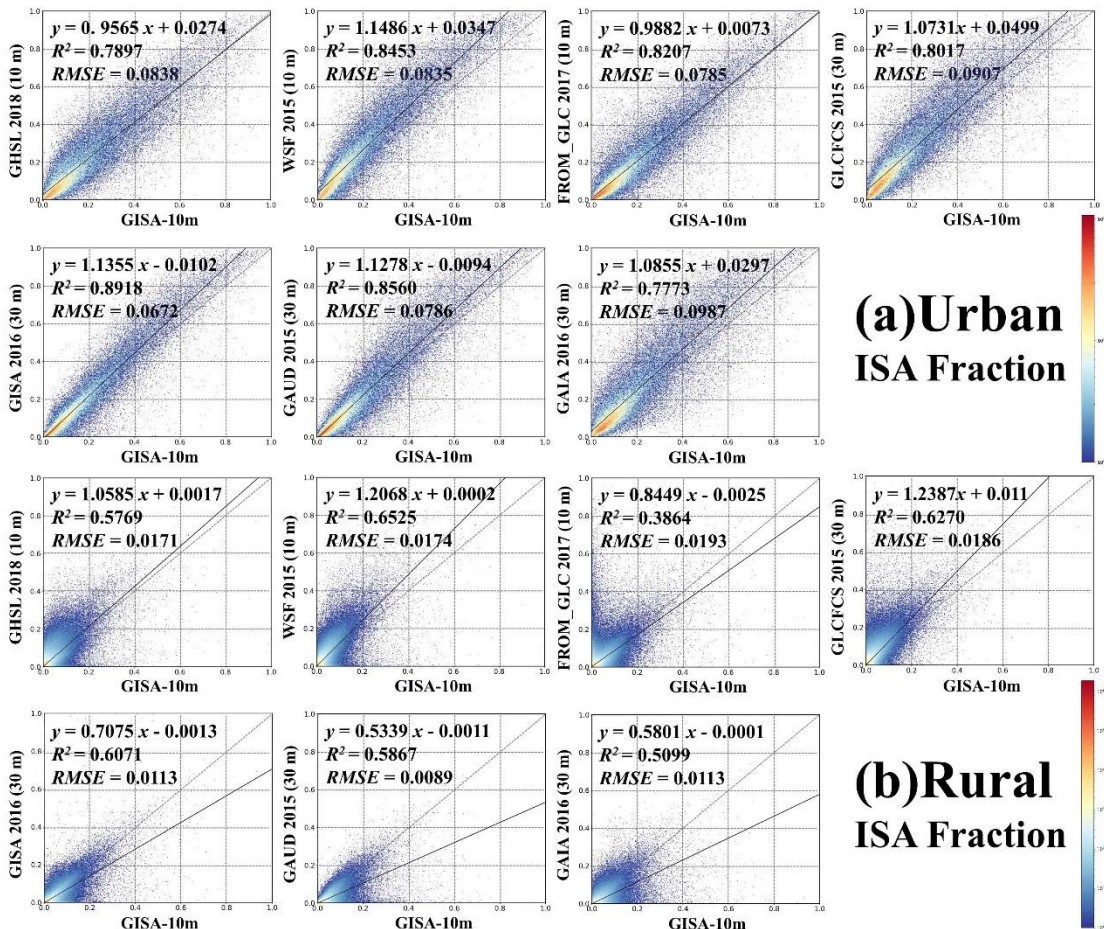

**Figure 11. Scatterplots of (a) urban and (b) rural ISA fraction between GISA-10m with GHSL, WSF, FROM_GLC10, GLCFCS, GAUD, GAIA, GISA, respectively. ISA fraction was calculated within a 0.05° by 0.05° spatial grid.**

In general, GISA-10m exhibited high agreement ($0.777 < R^2 < 0.892$) with these existing datasets over urban regions. In the

case of GHSL2018 and FROM_GLC10, their fitted lines with GISA-10m were closer to the 1:1 line in the high fraction

regions (Fig. 11a). As shown in the Fig.12, GHSL2018 and GISA-10m were generally similar in the dense urban areas (e.g.

urban cores in Fig.12), but GHSL2018 tended to overestimate ISA in the low-density residential areas (Fig.12c). The fitted

lines for GLCFCS and WSF2015 were above the diagonal (slope greater than 1 and intercept greater than 0) in both high and

low ISA fraction regions, possibly owing to their overestimations. For instance, in the case of Cairo (Fig. 12b), WSF2015

showed significant overestimations but other datasets better depicted the residential areas. According to Marconcini et al.,

(2020), the overestimations of the WSF2015 may be related to the employment of the coefficient of variation (COV), which

reduced the omissions in the rural regions but at the same time led to overestimations of ISA extent. The fitted lines for the

three 30-m datasets (i.e. GISA, GAIA, GAUD) were all above the diagonal (Fig. 11a), suggesting that they detected more

urban ISA than GISA-10 m. However, in the 30-m dataset, vegetation alongside roads or buildings was often identified as

ISA due to the issue of mixed pixels (Gong et al., 2020b). From this perspective, the results of GISA-10m seem more

reliable due to its high spatial resolution. For instance, in the case of Johannesburg and Los Angeles (Figs. 12c&d), GAIA and GAUD exhibited false alarms in both residential and industrial areas, but these errors were significantly reduced in GISA-10m due to the better discrimination ability of 10-m Sentinel data.

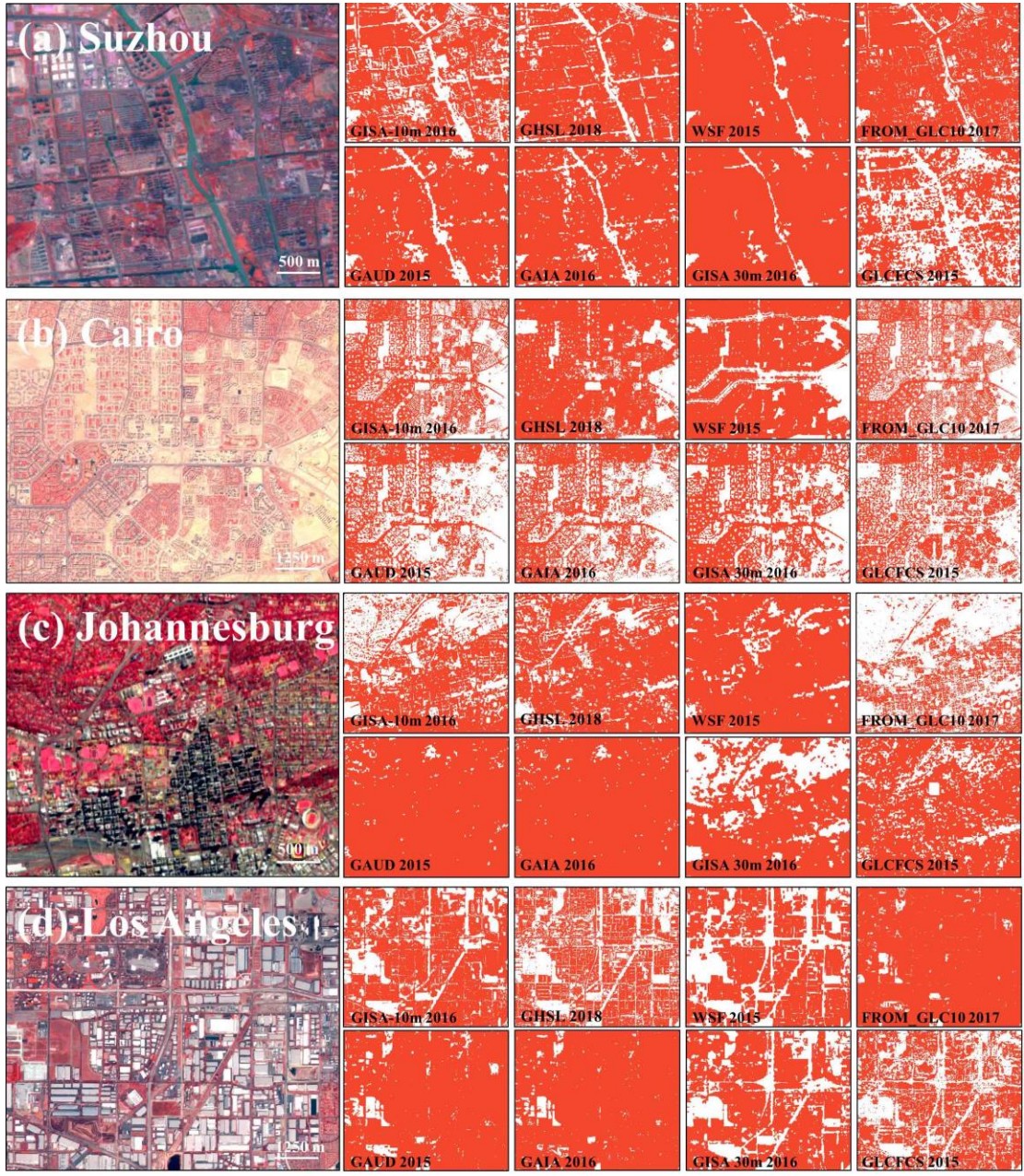

**Figure 12. Comparison of the GISA-10m and seven datasets over urban regions in (a) Suzhou, China; (b) Cairo, Egypt; (c) Johannesburg, South Africa; (d) Los Angeles, the United States. The Sentinel-2 images were composited in the false-color combination (R: NIR, G: Red, B: Green).**





On the other hand, the agreement between GISA-10m and existing datasets was slightly lower in rural regions ($0.5099 < R^2 < 0.6525$). The fitted slopes between three 30-m datasets (i.e. GISA, GAIA, GAUD) and GISA-10m in rural regions were all

less than one. This phenomenon can be attributed to the finer spatial resolution of GISA-10m, which detected more rural ISA than the 30-m datasets (Figs. 13b&d). As to GLCFCS and WSF2015, they possessed more rural ISA than GISA-10m (Fig. 11b), which may be attributed their overestimations. For example, in Figs. 13a&c, GLCFCS and WSF2015 failed to identify the vegetation in the village. FROM_GLC10 seemed more consistent with GISA-10m (see the sample of US, Fig. 13d), but it tended to underestimate the rural ISA (see Figs. 13a-c). GHSL2018 and GISA-10m showed high agreement in the rural

regions. However, GHSL2018 aimed to outline human settlements while GISA-10m focused on artificial ISA (including buildings, parking lot, roads).

Earth System Open Access
Science
Data Discussions

**Figure 13. Comparison of the GISA-10m and seven datasets over rural regions in (a) China (126.348044° E, 45.269079° N), (b) Uzbekistan (60.573313° E, 41.461425° N), (c) Côte d'Ivoire (5.853317° W, 6.820244° N), (d) the United States (90.210747° W, 39.950221° N). The illustration is of Sentinel-2 images with false-color combination (R: NIR, G: Red, B: Green) to enhance the ISA.**

The differences between GHSL2018, WSF2015 and GISA-10m were further analysed by taking Beijing and Washington as examples. In Fig. 14, the overlapping parts between these datasets were marked in different colors, and the regions where the three datasets all agreed were shown in gray. In both examples, WSF2015 and GHSL2018 tended to overestimate the ISA



extent (Fig. 14b). They wrongly identified vegetation as ISA in the low-density residential areas (Fig. 14h). In particular,
GHSL2018 successfully detected roads in Beijing but failed in Washington (see the color of purple in Fig. 14). This may be
related to the fact that GHSL2018 used different sources of training samples in different regions (Corbane et al., 2021).
Although WSF2015 generally obtained similar results with GISA-10m, its detected roads may stem from the overestimation
of building boundaries (Marconcini et al., 2020). For instance, WSF2015 ignored the airport runways in the example of
Beijing (Fig. 14d). In the case of Washington, WSF2015 was less capable of delineating scattered buildings than GISA-10m
and GHSL2018 (Fig. 14f), possibly because it also incorporated the 30-m Landsat data in the ISA detection. It should be
mentioned that GHSL2018 estimated the probability of human settlement, and hence, different thresholds could yield
different results. Small thresholds were suitable for capturing scattered settlements but could result in false alarms. In this
study, we chose 0.2 as the threshold, as suggested by Corbane et al., (2021).

**Figure 14. The illustration of WSF2015, GHSL2018 and GISA-10m in (i) Beijing and (ii) Washington. Regions where three datasets all agreed were shown in gray.**

## 5.2 Importance of multi-source features

In this paper, we proposed a global ISA mapping method that incorporated spectral, SAR, and temporal information extracted from multi-source Sentinel data. To illustrate the importance of multi-source features in the global ISA mapping,
we selected 30 hexagons in terms of the global urban ecoregions (Schneider et al., 2010). Urban ecoregions were defined with reference to biomes, urban landscapes, and economic levels. In each ecoregion, we randomly selected two grids, with their population greater or less than 5 million, respectively (Fig. S1). The "snow and ice" ecoregion was not considered. Feature contribution estimated by RF classifier was employed to analyze the relative importance of multi-source features (Pflugmacher et al., 2014; Zhang et al., 2020a). Different color schemes in Fig. 15 indicated different types of features. For
instance, the color of blue denoted SAR features while green represented the spectral indices. The results indicated that the feature importance varied in different regions. For example, SAR features were more effective in the temperate grassland of Middle East and Asia (53N_75E and 50N_39E), while phenological features had more influence in the deciduous forest of Siberia (65N_125E). In particular, SAR features played a more important role in the more populated regions, e.g. in temperate forest of North America and Europe as well as temperate grassland of the Middle East and Asia (Fig. 15).

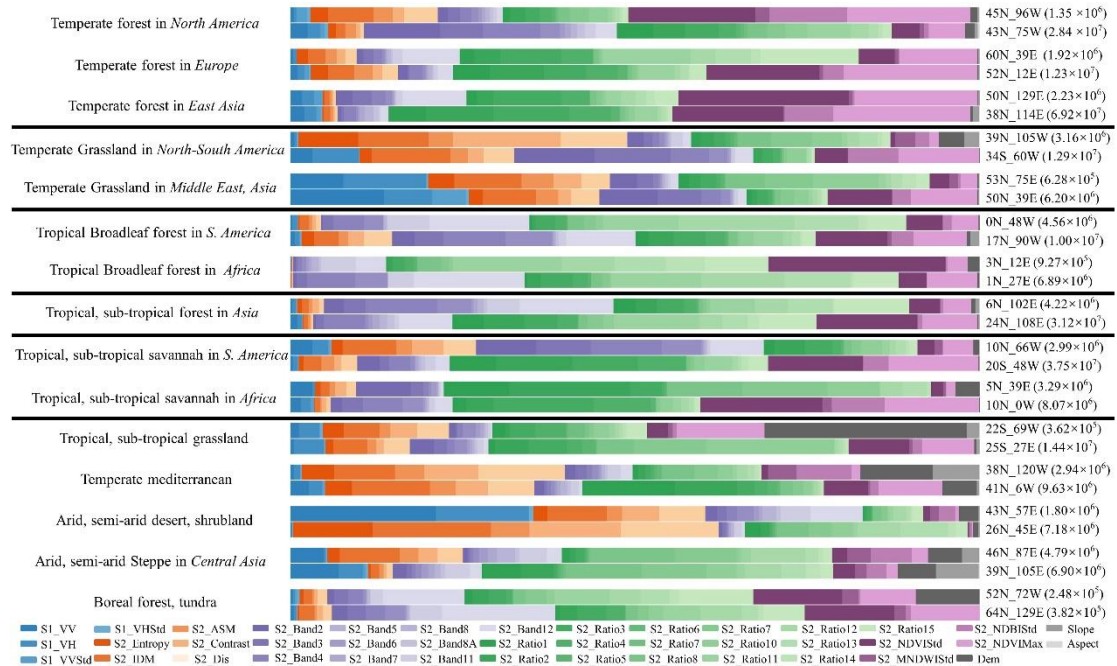


**Figure 15. Relative importance of multi-source features in the 30 randomly selected grids located in different urban ecoregion. The labels on the right denote grid ID and total population. The Dis, IDM and ASM represents the dissimilarity, angular second moment and inverse difference moment, respectively.**

It is worth noting that although high-rise ISA (e.g. buildings) tended to have higher radar backscatters, the importance of
SAR features was not always the highest. For example, in the hexagon of central US (45N_96W), SAR features played a less significant role than temporal metrics. In contrast, the spectral indices and phenological information were more effective in



this region. For example, as shown in Fig. 16 (red squares), in the residential area, the buildings were often surrounded by dense shrub, which may shrink the double bounce scattering. Therefore, spectral and phenological features had higher importance since they can better distinguish vegetation from non-vegetation. A similar situation occurred in a desert area
(26N_45E), where SAR features could not distinguish well ISA from NISA due to the complex topography of mountains. In this case, spectral indices and textures were more effective (Fig. 15). However, SAR features were still very important for global ISA mapping, especially for identifying rural buildings (Zhang et al., 2020a). Therefore, in this study, we used multi-source features and hexagon-based adaptive random forest models to ensure that the most suitable features were chosen for different regions.

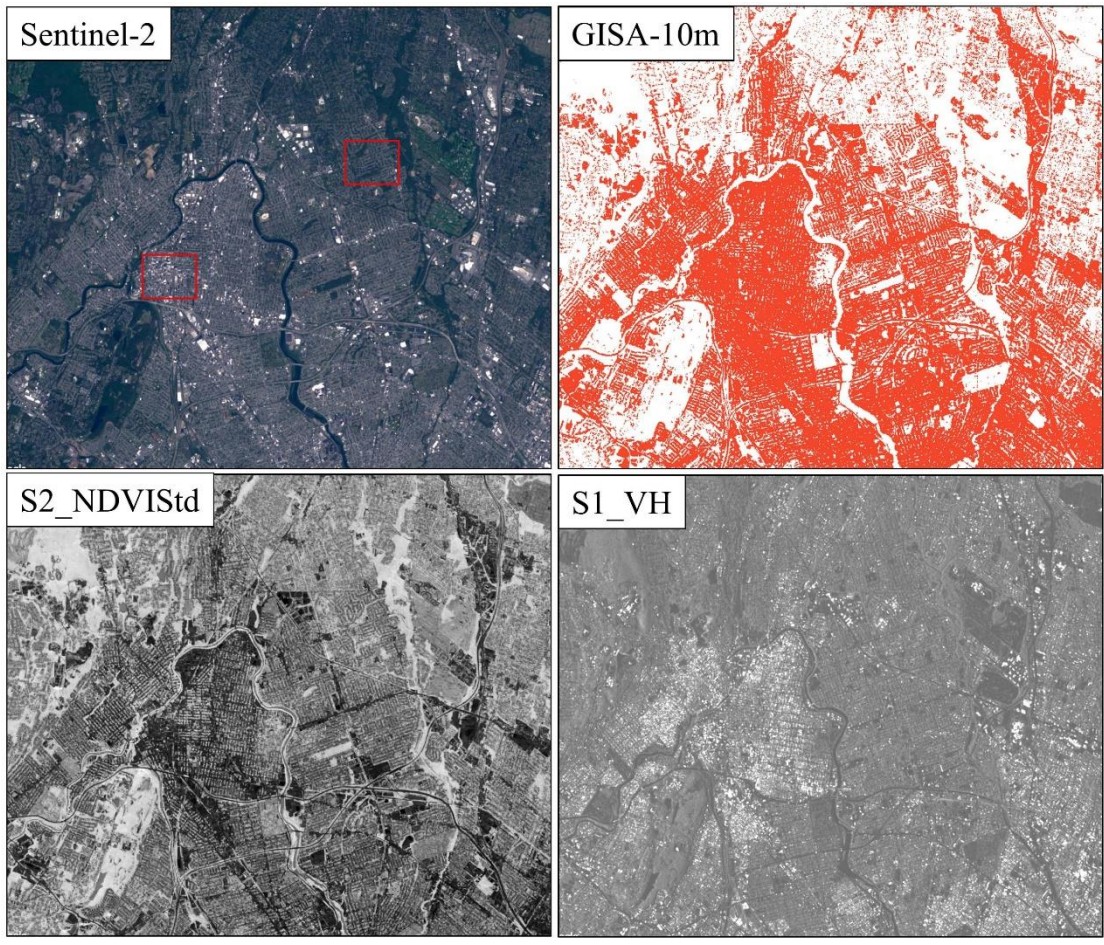


**Figure 16. Example of Sentinel-1 VH backscatter, standard deviation of NDVI from Sentinel-2 (S2_NDVIStd), Sentinel-2 true color composite and GISA-10m in Paterson, New Jersey, the United States.**

### 5.3 Impact of training sample size

Based on aforementioned randomly selected 30 hexagons in different urban ecoregions, we investigated the relationship
between the training sample size and the accuracy (Fig. S1). For each hexagon, we fixed the number of NISA samples to

30,000 and changed the number of $ISA_{RS}$ and $ISA_{OSM}$ samples. Specifically, we first randomly selected 1,000 $ISA_{RS}$, 1,000 $ISA_{OSM}$ and 2,000 NISA samples from the candidate pool (see Section 3.1.1) as the test samples and used the remaining ones for the training. We randomly selected 50 $ISA_{RS}$ and 50 $ISA_{OSM}$ as the initial training samples, and subsequently, in an iterative manner, 400 $ISA_{RS}$ and $ISA_{OSM}$ samples were randomly selected from the pool and added to the training samples to

train the RF classifier. It can be observed that all the hexagons reached saturation with 2,500 $ISA_{RS}$ and $ISA_{OSM}$ samples (Fig. 17). Therefore, in this research, we set the number of $ISA_{RS}$, $ISA_{OSM}$, and NISA samples to 2,500, 2,500 and 30,000, respectively.



**Figure 17. The F-Score as a function of $ISA_{RS}$ and $ISA_{OSM}$ samples in the randomly selected 30 global grids.**





## 5.4 Advantages of locally adaptive RF classification

We used two hexagons located in China (CHN) and Saudi Arabia (SA) to demonstrate the advantages of adaptive random forest classification. Although China and Saudi Arabia are both located in Asia, their urban landscapes and architecture styles are significantly different due to their difference in climate, environment, and culture. In this section, we migrated the training samples from one hexagon to classify the other one. For example, training samples collected in the SA was used to classify the hexagon of China. The accuracy of each hexagon was evaluated by the visually-interpreted samples inside it. It was found that the OA decreased by 34% when the SA samples was applied to CHN (written as SA-to-CHN). Similarly, the OA was substantially reduced by 23% by the transfer of CHN-to-SA. Furthermore, we found that local samples always outperformed the migrated ones (see Table 8), which verified the necessity of local and adaptive classification strategies in the global ISA mapping. Besides, the locally adaptive model is more sensitive to the sample quality compared to the global model (Radoux et al., 2014), which further showed the necessity and effectiveness of the local classification strategy.

**Table 8. Results of quantitative accuracy assessment for China (CHN) and Saudi Arabia (SA) based on local and transferred samples. OA denotes the overall accuracy.**

| | Saudi Arabia | | | | China | | | |
|---|---|---|---|---|---|---|---|---|
| | OA (%) | Kappa | F-Score of ISA (%) | F-Score of NISA (%) | OA (%) | Kappa | F-Score of ISA (%) | F-Score of NISA (%) |
| ISA_SA & NISA_SA | **93.00** | **0.8599** | **92.39** | **93.95** | 79.50 | 0.5915 | 77.60 | 81.86 |
| ISA_SA & NISA_CN | 53.00 | 0.7253 | 65.44 | 26.77 | 55.00 | 0.5233 | 4.35 | 70.59 |
| ISA_CN & NISA_SA | 70.50 | 0.8396 | 53.23 | 78.55 | 48.00 | 0.6251 | 63.38 | 10.53 |
| ISA_CN & NISA_CN | 50.50 | 0.0846 | 64.77 | 16.95 | **89.00** | **0.7778** | **86.90** | **91.30** |

## 5.5 Influence of the sources of training samples

In this section, the effects of the training sample sources, i.e., from remote sensing dataset ($ISA_{RS}$) and the OSM ($ISA_{OSM}$), were investigated. Various combinations of the $ISA_{RS}$ and $ISA_{OSM}$ samples were tested at the global scale (Table 9). In general, it can be found that using both sources yielded the most accurate results, which showed the effectiveness and necessity of incorporation of training samples from remote sensing and crowdsourcing OSM. By further checking the UA and PA of ISA, one can see that using both sample sources significantly improved the PA and reduced the ISA omissions, since the combination of $ISA_{RS}$ and $ISA_{OSM}$ strengthened the diversity of the training samples. Similarly, it was also found that the multi-source samples significantly raised the PA of NISA and lowered its commission error. Although the quality and consistency of OSM data may affect the performance of GISA-10m (Fan et al., 2014; Zacharopoulou et al., 2021), global ISA mapping using only $ISA_{OSM}$ showed relatively consistent accuracy across continents (Fig.S3). This can be attributed to rule-based method we implemented that improved the reliability and spatial consistency of $ISA_{OSM}$. In addition, the collaboration of $ISA_{OSM}$ improved the overall accuracy of global ISA mapping by 3% (Table 9), indicating the feasibility of OSM data in enhancing performance of global ISA mapping after a set of refinements.

Earth System
Science
Data

**Table 9. Results of global accuracy assessment for ISA$_{RS}$ and ISA$_{OSM}$ sample. OA denotes the overall accuracy, while PA and UA indicate the user's accuracy and the producer's accuracy, respectively.**

| Source of training sample | OA (%) | Kappa | F-Score of ISA (%) | F-Score of NISA (%) | UA of ISA (%) | PA of ISA (%) | UA of NISA (%) | PA of NISA (%) |
|---|---|---|---|---|---|---|---|---|
| NISA+ISA$_{RS}$+ISA$_{OSM}$ | **86.06** | **0.7165** | **83.65** | **88.55** | 86.13 | **81.30** | **86.01** | 91.25 |
| NISA+ISA$_{RS}$ | 80.24 | 0.5871 | 73.85 | 84.63 | **88.16** | 63.54 | 76.73 | **94.35** |
| NISA+ISA$_{OSM}$ | 82.99 | 0.6500 | 78.96 | 86.34 | 86.24 | 72.81 | 81.17 | 92.23 |

## 6 Data availability

The GISA-10m product generated in this study is available in the public domain at http://doi.org/10.5281/zenodo.5791855
(Huang et al, 2021). Sentinel data were acquired from the Google Earth Engine (available at code.earthengine.google.com,
last access: 6 August 2021). GHSL was provided by the Joint Research Centre at European Commission (available at
https://ghsl.jrc.ec.europa.eu/datasets.php, last access: 19 December 2021). WSF was provided by the German Aerospace
Center (https://doi.org/10.6084/m9.figshare.c.4712852, Marconcini et al., 2020).The GlobeLand30 and GAUD were
downloaded from the website of the National Geomatics Center of China (available at http://www.globallandcover.com/, last
access: 6 August 2021) and Sun Yat-sen University (available at https://doi.org/10.6084/m9.figshare.11513178.v1, Liu et al.,
2020b). FROM_GLC10 and GAIA were assessed from the Tsinghua University (available at http://data.ess.tsinghua.edu.cn,
last access: 6 August 2021). The GISA was provided by the Institute of Remote Sensing Information Processing at Wuhan
University (available at https://zenodo.org/record/5136330, Huang et al., 2021a). GLCFCS was provided by Aerospace
Information Research Institute at Chinese Academy of Sciences (available at https://zenodo.org/record/4280923, Zhang et al.,
2020b). The planet files were download from the website of OpenStreetMap (available at https://planet.openstreetmap.org,
last access: 19 December 2021).

## 7 Conclusion

In this study, we proposed a global ISA mapping method and produced the 10-m global ISA dataset (GISA-10m). To our
knowledge, this is the first global 10-m ISA map based on Sentienl-1 and 2 data. To this end, a global training sample
generation method was proposed based on a series of temporal, spatial, spectral, and geometrical rules and 58 million
training samples were generated from the existing global ISA datasets and the social sensing data (i.e., OSM). On the basis
of the 2.7 million Sentinel images available in the Google Earth Engine (GEE), multi-source features were constructed
including spectral, texture, SAR, and temporal metrics. The global terrestrial surface was divided with hexagons, and the
results were obtained by a series of RF classifiers. In particular, the mapping was conducted adaptively for each hexagon, by
considering the difficulty and diversity for the global ISA detection. The overall accuracy of GISA-10m exceeded 86%
based on a set of independent test samples. The inter-comparison between different global ISA datasets showed the

superiority of our results. Based on GISA-10m, the ISA distribution at the global, continental, and country levels was discussed and compared. In addition, the global ISA distribution was compared between rural and urban. In particular, for the first time, by courtesy of the high spatial resolution, the global road ISA was further identified and its distribution was discussed.

GISA-10m can be used for global climate change studies and urban planning. Our proposed rule-based sample generation method can also be applied for global mapping of other land cover categories. For example, millions of cropland and forest tags in the OSM can facilitate global high-resolution cropland and forest mapping. The ISA mapping method via multi-source geospatial data presented in this paper can also be improved by incorporating additional data sources, such as building footprints from Microsoft and Facebook (Corbane et al., 2021). In the future, we plan to extend the temporal coverage of GISA-10m and reveal the global ISA dynamics at the 10-m resolution.

**Author contributions.** XH conceived the study. XH, JY, WW and ZL designed and implemented the methodology. JY prepared original draft and XH revised it.

**Competing interests.** The authors declare that they have no conflict of interest.

**Financial support.** The research was supported by the National Natural Science Foundation of China under Grant 41971295, and the Foundation for Innovative Research Groups of the Natural Science Foundation of Hubei Province under Grant 2020CFA003.

**Acknowledgments.** The authors greatly appreciate the free access of the Sentinel data provided by the ESA, the GlobaLand30 provided by the National Geomatics Center of China, the FROM_GLC10 provided by Tsinghua University, and the GISA data provided by Wuhan University. We thank the Google Earth Engine team for their excellent work to maintain the planetary-scale geospatial cloud platform, as well as volunteers around the world contributed to the OpenStreetMap.

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
