# Peer review of "Mapping 10-m global impervious surface area (GISA-10m) using multi-source geospatial data"

_Earth System Science Data, 2021_

## Author Comment (AC1)

Accurate mapping of artifical impervious surface using remote sensing is challenging, especially at continental and global scales. The authors here provides an really exciting ISA map dataset which makes the above-mentioned challenge partially addressed. Given a 10 m spatial resolution, the GISA-10m is able to detect some subtle patterns that cannot be extracted by previously 30~300m products. As a public user, I only have one concern as follows.

The city group often include multiple cities with different scales. So, I believe it is important for potential users to know the accuracy of the GISA-10m for the cities with different scales, i.e., small, middle, and big city. Is it possible to compare the overall accuracy of the GISA-10m across different city sizes?

R: Thanks for your comment. We extracted the visually-interpreted samples located in cities and divided them into three levels (i.e., small, middle and big cities) to assess the accuracy of GISA-10m over cities with different scales: Level 1 (population<250,000), Level 2 (250,000 to 1,000,000), and Level 3 (>1,000,000) (Larkin et al., 2016; Yang et al., 2019). It was found that the overall accuracy of GISA-10m across three levels of cities was 85.35%, 87.43% and 85.42%, respectively (Table R1). The result indicated the performance of GISA-10m in different scales of cities was stable, and was also close to its global assessment (OA of 86.06%).

**Table R1**. Results of quantitative accuracy assessment for three level of cities: Level 1 (population<250,000), Level 2 (250,000 to 1,000,000), and Level 3 (>1,000,000). OA represents the overall accuracy.

| Level of cities | OA (%) | Kappa | F-Score of ISA (%) | F-Score of NISA (%) |
|---|---|---|---|---|
| Level 1 | 85.35 | 0.2205 | 91.92 | 30.41 |
| Level 2 | 87.43 | 0.2189 | 93.11 | 29.41 |
| Level 3 | 85.42 | 0.4005 | 91.86 | 47.06 |

Reference:

Characteristics and Air Quality in East Asia from 2000 to 2010, Environ. Sci. Technol., 50(17), 9142–9149, doi:10.1021/acs.est.6b02549, 2016.

Yang, Q., Huang, X. and Tang, Q.: The footprint of urban heat island effect in 302 Chinese cities: Temporal trends and associated factors, Sci. Total Environ., 655, 652–662, doi:10.1016/j.scitotenv.2018.11.171, 2019.

---

## Author Comment (AC2)

This manuscript proposed an efficient method to produce the 10-m global impervious surface areas (GISA-10m) based on the existing ISA maps, Sentinel-1/2 images, and OSM data. Compared to existing global GISA products, GISA-10m can provide higher spatial resolution while keeping higher accuracy. The inter-comparison with existing datasets demonstrated the superiority of GISA-10m. Analysis of ISA on rural and urban areas further revealed the urbanization level and landscape of different countries in more details. In particular, an interesting point of GISA-10m is that it is able to delineate the area of roads across the world, making GISA-10m valuable for relevant urban studies. In general, this manuscript is well presented and makes novel contributions. However, some issues should be clarified to improve this manuscript. Specific comments include the following aspects:

1) In Section 3.1.3, the authors used 200 trees for training the random forest classifier, while the effect of the number of trees is not analyzed. Besides, the key parameter, e.g., the number of features used for training each tree, is not clarified. Please provide this information for better understanding.

R: Thanks for your comments. We analyzed the effect of the number of trees on the accuracy of global ISA mapping, using 30 mapping grids (hexagons with sides of two degrees) from global urban ecological regions (see Section 5.2 for details). The results showed that the overall accuracy was low and unstable while the number of trees was less than 20 (Fig. R1). As the number of trees increased, the mapping accuracy increased and stabilized around 200 trees. Therefore, we used 200 trees for each random forest model in GISA-10m.

In terms of the features used to train each tree, the random forest uses a random subset of features to reduce the correlation between trees. In general, the diversity of trees can be increased when fewer features are used for training each tree (Breiman, 2001). In GISA-10m mapping, we set the number of features used for each tree to the square root of the total number of features, as suggested by Liu et al., (2020).

[Figure]

**Figure R1**. The overall accuracy as a function of number of trees.

Reference:

Breiman, L.: Random forests, Mach. Learn., 45(1), 5–32, doi:10.1023/A:1010933404324, 2001.

Liu, H., Gong, P., Wang, J., Clinton, N., Bai, Y. and Liang, S.: Annual dynamics of global land cover and its longterm changes from 1982 to 2015, Earth Syst. Sci. Data, 12(2), 1217–1243, doi:10.5194/essd-12-1217-2020, 2020.

2) Line 90: do you mean by "operating by"?

R: Corrected.

3) Line 103: relevant reference should be provided to support "the terrain distortion caused by the combination of two orbits".

R: Added.

4) L120: it should be "Landsat 8".

R: Corrected.

5) L121: I found both "GLCFCS" and "GLC_FCS" in the manuscript. Please explain.

R: Thanks for pointing out this issue. "GLCFCS" and "GLC_FCS" both refer to the Global Land Cover with Fine Classification System generated by Zhang et al (2021). We have checked it throughout the manuscript.

Reference:

Zhang, X., Liu, L., Chen, X., Gao, Y., Xie, S. and Mi, J.: GLC_FCS30: global land-cover product with fine classification system at 30 m using time-series Landsat imagery, Earth Syst. Sci. Data, 13(6), 2753–2776, doi:10.5194/essd-13-2753-2021, 2021.

6) Line 201: the original OSM data are provided in vector form. When this data was converted to 10-m raster, whether the majority rule was applied? The majority rule refers to "a pixel (10m × 10m) was labelled as ISA if more than half of its area was cover by ISA, otherwise it was identified as NISA". Please clarified this issue.

R: Thanks for your comment. Usually, we have to rasterize the raw vector data into a higher resolution (i.e., less than 10 meter), before the majority rule can be applied. This was extremely time-consuming and computationally intensive when it is applied to global ISA mapping at 10-m. Therefore, we extracted the geometric center of a vector as the sample point, rather than converted it to a raster. In such way, the amount of training samples can be guaranteed while the computational cost was reduced. Moreover, we removed buildings with area less than 100 $m^2$ (~ a Sentinel pixel) to ensure the reliability of the sample, since the training sample extracted from the geometric center may be NISA (Non-ISA), when the area of a building is smaller than a Sentinel pixel.

7) L295: why the total number of visually interpreted samples was 10800 when 200 samples were selected in 59 grids? Please check.

R: Thank you pointing out this issue. It should be 11,800.

8) Section 3.3: it is better to move this section to Section 5.1, since the detailed discussion has been presented in Section 5.1.

R: Thanks for your suggestion. Section 3.3 has been moved to Section 5.1.

9) Figure 6: is it possible to compare the continental accuracies of other datasets presented in Table 3? The comparison at continental level may give a clear difference of different datasets.

R: Much obliged. We compared the overall accuracy of different datasets across continents. The results showed that the average overall accuracy of GISA-10m is more stable across six continents, and exceeded the existing datasets in Africa, North America and Europe. In addition, it was found that the performance of GHSL2018 and GLCFCS was relatively unstable in South America and North America, respectively.

[Figure]

**Figure R2**. Box plots of overall accuracy for GISA-10m and existing datasets in the six continents.

R: According to your comment, Figure 8 has been moved to the supplements.

11) Figure 11: it is better to use (a), (b), (c) to distinguish each subgraph.

R: Thanks for your suggestion. Label has been added to each subgraph (Fig. R3).

[Figure]

**Figure R3.** Scatterplots of urban and rural ISA fraction between GISA-10m with GHSL, WSF, FROM_GLC10, GLCFCS, GAUD, GAIA, GISA, respectively. ISA fraction was calculated within a 0.05° by 0.05° spatial grid.

12) Figure 17: this figure is not clear enough for presenting 30 grids. It is suggested to add legend and put this figure to the supplementary materials.

R: Thanks for your suggestion. We have added a legend to the figure (Fig. R4) and put it in the supplementary materials.

[Figure]

**Figure R4**. The F1-Score as a function of ISARS and ISAOSM samples in the randomly selected 30 global grids.

13) Line 372: "extracted" or "detected"?

R: Corrected.

14) Table 9: whether test samples used in Table 9 are from visually interpreted samples? Please clarify this.

R: Thank you for your comments. The test samples used in the Table 9 were the visually-interpreted samples described in Section 3.2. Accordingly, the relevant statement has been revised as: "*Various combinations of the ISARS and ISAOSM training samples were tested at the global scale using the visually-interpreted samples from Section 3.2 (Table 9)*"

---

## Author Comment (AC3)

The mapping of 10-m impervious surfaces at the global scale using multiple geodata sources is interesting. The authors applied temporal-spatial-spectral-geometrical rules to generate samples, and validation of the results is comprehensive and adequate. They also attempted to delineate the spatial distribution of impervious surface in urban and non-urban areas. The manuscript fits the journal's scope and the dataset is valuable, which is suitable for publication in ESSD. However, the paper still has some flaws (see my comments below) which should be further clarified or discussed before acceptance.

My major concern lies in the completeness and correctness of the OSM data. How about the effect of the geographic bias in spatial distribution of OSM data? More analysis is needed to discuss this issue.

R: Thank you for your comments. Given that geographic bias in the spatial distribution of OSM data may affect the mapping results, we applied temporal and spatial rules to mitigate the effect of the difference of the spatial distribution. In addition, spectral rule was used to remove potential errors in OSM-derived training samples (i.e., $ISA_{OSM}$). In fact, more than 82% of OSM ways are buildings and highways, whose total number exceeds 700 million (https://taginfo.openstreetmap.org/keys, last access: 20 June 2022). Therefore, OSM data provides a potential reference for large-scale ISA mapping, but it has rarely been employed in global ISA mapping. According to your comments, we calculated the overall accuracy for the test grids where the number of $ISA_{OSM}$ training samples were less or larger than 2500 (i.e., the recommended size of training sample in Section 5.3). The results showed that the accuracy of these regions was similar to the global accuracy (Table R1). This phenomenon demonstrated the stable performance of GISA-10m. Moreover, global ISA mapping involved only $ISA_{OSM}$ showed relatively stable accuracy across the continents (Fig. R1), suggesting that the refined OSM buildings and roads can reduce the impact of their uneven spatial distribution. Overall, although the spatial distribution of OSM data is uneven, we tried to balance its spatial distribution through a series of rules, and incorporated multi-source geospatial data (e.g., satellite-derived datasets) to reduce the impact of geographical bias on GISA-10m.

**Table R1**. Results of quantitative accuracy assessment for test grids with $ISA_{osm}$ less or more than the recommended size via visually-interpreted samples. OA represents the overall accuracy.

| Type of test grids | OA (%) | Kappa | F-Score of ISA (%) | F-Score of NISA (%) |
|---|---|---|---|---|
| $ISA_{osm}$ less than recommended size | 85.61 | 0.7021 | 81.79 | 89.01 |
| $ISA_{osm}$ more than recommended size | 86.23 | 0.7218 | 84.32 | 88.35 |
| All of the above | 86.06 | 0.7165 | 83.65 | 88.55 |

[Figure]

**Figure R1**. Box plots of the overall accuracy for GISA-10m in the six continents by using ISA$_{OSM}$.

Line 10: "global ISA mapping" should be "global ISA datasets"

R: Corrected.

Line 21: "refined OSM data" -> "OSM data".

R: Done.

Line 80-85: The GISA-10m dataset attempted to further delineate road regions from the ISA. This should be mentioned in the introduction and abstract.

R: Much obliged. Done.

Line 152: "multiple sources" is not clear, and can be modified as "multi-source datasets".

R: Done.

Figure 1. It would be better to label each step, e.g., "Step 1. Training sample generation".

R: Done.

Line 157. The authors selected the GlobeLand30 in 2010 but chosed other data (e.g., GISA and FROM-GLC) in 2016. Would the temporal gap between these data impact the quality of training data?

R: Thanks for your comments. In GlobeLand30, extensive visual interpretation was employed to detect artificial surfaces. Therefore, it was used in our study to effectively reduce false alarms from other datasets (i.e., GISA and FROM_GLC10) (Chen et al., 2015). Although there is a six-year gap between GlobeLand30 and other datasets, we adopted the commonly used assumption that the transition from ISA to NISA rarely happened (Gong et al., 2020; Huang et al., 2021, 2022; Li and Gong, 2016), so that GlobeLand30 in 2010 can be used for GISA-10m mapping.

Reference:

Chen, J., Chen, J., Liao, A., Cao, X., Chen, L., Chen, X., He, C., Han, G., Peng, S., Lu, M., Zhang, W., Tong, X. and Mills, J.: Global land cover mapping at 30 m resolution: A POK-based operational approach, ISPRS J. Photogramm. Remote Sens., 103, 7–27, doi:10.1016/j.isprsjprs.2014.09.002, 2015.

Gong, P., Li, X., Wang, J., Bai, Y., Chen, B., Hu, T., Liu, X., Xu, B., Yang, J., Zhang, W. and Zhou, Y.: Annual maps of global artificial impervious area (GAIA) between 1985 and 2018, Remote Sens. Environ., 236, 111510, doi:10.1016/j.rse.2019.111510, 2020.

Huang, X., Li, J., Yang, J., Zhang, Z., Li, D., Liu, X., Xin, H., Jiayi, L., Jie, Y., Zhen, Z., Dongrui, L. and Xiaoping, L.: 30 m global impervious surface area dynamics and urban expansion pattern observed by Landsat satellites: From 1972 to 2019, Sci. CHINA Earth Sci., doi:10.1007/s11430-020-9797-9, 2021.

Huang, X., Song, Y., Yang, J., Wang, W., Ren, H., Dong, M., Feng, Y., Yin, H. and Li, J.: Toward accurate mapping of 30-m time-series global impervious surface area (GISA), Int. J. Appl. Earth Obs. Geoinf., 109, 102787, doi:https://doi.org/10.1016/j.jag.2022.102787, 2022.

Li, X. and Gong, P.: An "exclusion-inclusion" framework for extracting human settlements in rapidly developing regions of China from Landsat images, Remote Sens. Environ., 186, 286–296, doi:https://doi.org/10.1016/j.rse.2016.08.029, 2016.

Line 171. How did you define edge pixels? I think the edge pixels are different between 30-m and 10-m images, as a non-edge pixel in a 30m image may be edge pixels in a 10m image. Could you clarify this issue?

R: Thanks for your comments. Edge pixels were defined as the outermost pixels of each ISA patch. We removed the edge pixels in each dataset, and then selected their ISA intersection as potential training samples. In this way, errors contained in non-edge pixels in the 30-m data (e.g., mixed pixels) can be removed by the edge pixels in the 10-m data. Moreover, we further applied the spectral rules to remove the erroneous samples.

Line 197. Why buildings with area less than 100 m2 were excluded?

R: Thank you for your comments. We removed buildings with area less than 100 $m^2$ (~ a Sentinel pixel) to ensure the reliability of the sample. Because the training sample extracted from the geometric center may be NISA (Non-ISA), when the area of a building is smaller than a Sentinel pixel.

Line 210. Why did the authors remove the OSM samples intersected with those from other global datasets?

R: Thanks for your comment. In the field of supervised classification, the diversity of samples was important for the generalization ability of the classification model (Huang and Zhang, 2013). Considering that $ISA_{OSM}$ could overlie with $ISA_{RS}$, we removed the $ISA_{OSM}$ samples intersected with $ISA_{RS}$ sample pool to increase the diversity and reduce the redundancy of the ISA samples.

Reference:

Huang, X., Zhang, L., 2013. An SVM Ensemble Approach Combining Spectral, Structural, and Semantic Features for the Classification of High-Resolution Remotely Sensed Imagery. IEEE Trans. Geosci. Remote Sens. 51, 257–272. https://doi.org/10.1109/TGRS.2012.2202912

Line 235. Please explain why these features were chosen.

R: Thank you for your comments. A total of 41 features were built for GISA-10m mapping in terms of spectrum, texture, phenology, SAR, and topography. Firstly, we used the spectral signatures provided by Sentienl-2 data to extract ISA in visible, red-edge, near-infrared and infrared bands. In addition, considering that spectral indices could increase the differences between land covers, we also extracted a series of normalized spectral indices to enhance the discrimination ability between ISA and NISA (Yang and Huang, 2021). The complex spectral and spatial characteristics in urban environments increase the difficulty of ISA mapping. In this regard, texture features are usually employed to depict the spatial information of urban ISA (Huang and Zhang, 2013). Therefore, we extracted GLCM textures to describe the spatial patterns of ISA. SAR data is potential for reducing the false alarms caused by bare soil in optical images, and it is more sensitive to buildings. In addition, it is able to penetrate clouds. So, in this study, it was combined with optical data for ISA mapping. Given that spectra and backscatter of some NISA (e.g., vegetation and water bodies) vary throughout time, the phenological information derived from multi-temporal spectral and SAR data is utilized to depict the temporal fluctuations. Topography-related features are necessary for ISA mapping, in order to reduce the confusion between complex terrain and buildings. For instance, topographical features could help to distinguish steeply hills from buildings.

Reference:

Huang, X. and Zhang, L.: An SVM Ensemble Approach Combining Spectral, Structural, and Semantic Features for the Classification of High-Resolution Remotely Sensed Imagery, IEEE Trans. Geosci. Remote Sens., 51(1), 257–272, doi:10.1109/TGRS.2012.2202912, 2013.

Yang, J. and Huang, X.: The 30 m annual land cover dataset and its dynamics in China from 1990 to 2019, Earth Syst. Sci. Data, 13(8), 3907–3925, doi:10.5194/essd-13-3907-2021, 2021.

Line 286. How many RF models were built?

R: Thanks for your comment. We divided the global terrestrial surface using 1,808 hexagons where a local RF model was built for adaptive ISA classification in each hexagon. Therefore, a total of 1,808 RF models were built.

Line 293. How did the authors select the ISA test points? If the points were mostly located in urban areas, it might bias the assessment result. Could you provide the ISA density around these ISA points?

R: Thanks for your comments. The cluster sampling was used to determine the location of test samples (Stehman and Foody, 2019). Specifically, 59 grids (1°×1°) were first randomly chosen across six continents based on population, ecoregion, and urban landscape. The ISA test samples were then obtained in each grid by random sampling and visual interpretation from high-resolution Google Earth images. In such way, samples from different urban sizes and densities were considered for validation. According to your suggestion, we provided the ISA density around the ISA test samples (0.5 km buffer). As seen from Fig. R2, the test samples involved not only high-density ISA samples in urban areas, but also a large number of low-density samples in suburban and rural regions. According to your suggestion, we have added this figure in the Figure 5.

[Figure]

**Figure R2**. ISA density for ISA test samples.

Reference:

Stehman, S. V and Foody, G. M.: Key issues in rigorous accuracy assessment of land cover products, Remote Sens. Environ., 231, 111199, doi:https://doi.org/10.1016/j.rse.2019.05.018, 2019.

Figure 9. It's interesting to see the accuracy in rural and arid areas. How about urban areas?

R: Thanks for your comment. In the case of urban region, GISA-10m exhibited satisfactory result with an overall accuracy similar to the global assessment (Table R2). Note that urban ISA only accounts for one-third of global ISA while nearly 70% of ISA was located in suburban and rural regions. Existing datasets showed relatively more ISA omissions in rural or arid regions, suggesting that global ISA mapping at 10-m (e.g., GISA-10m) is necessary. Moreover, we divided the visually-interpreted samples located in cities into three levels (i.e., small, middle and big cities) to assess the accuracy of GISA-10m over cities with different scales: Level 1 (population<250,000), Level 2 (250,000 to 1,000,000), and Level 3 (>1,000,000) (Larkin et al., 2016; Yang et al., 2019). It was found that the overall accuracy of GISA-10m across three level of cities was 85.35%, 87.43% and 85.42%, respectively (Table R3). The result indicated the performance of GISA-10m in different scales of cities was stable, and was also close to its global assessment (OA of 86.06%).

**Table R2**. Results of quantitative accuracy assessment via visually-interpreted and ZY-3 samples in urban regions between GISA-10m and the existing ISA datasets. OA represents the overall accuracy.

| Urban Regions | Visually interpreted samples (n=2253) | | | | ZY-3 samples (n=24418) | | | |
|---|---|---|---|---|---|---|---|---|
| | OA (%) | Kappa | F-Score of ISA (%) | F-Score of NISA (%) | OA (%) | Kappa | F-Score of ISA (%) | F-Score of NISA (%) |
| GISA-10m | **85.49** | **0.30** | **91.93** | **38.26** | 77.96 | **0.52** | 82.71 | **69.61** |
| GHSL2018 | 76.61 | 0.20 | 86.02 | 31.41 | 76.56 | 0.47 | 82.38 | 64.99 |
| GLCFCS | 78.43 | 0.18 | 87.51 | 27.96 | 75.75 | 0.48 | 80.98 | 66.55 |
| WSF2015 | 83.58 | 0.23 | 90.73 | 32.76 | **78.36** | 0.49 | **84.64** | 63.38 |
| FROM_GLC10 | 75.32 | 0.21 | 85.15 | 31.66 | 74.78 | 0.45 | 80.35 | 64.80 |
| GISA | 82.96 | 0.24 | 90.41 | 33.15 | 78.09 | 0.49 | 84.25 | 63.98 |
| GAUD | 81.49 | 0.22 | 89.49 | 31.06 | 78.20 | 0.50 | 84.07 | 65.48 |
| GAIA | 84.02 | 0.20 | 91.07 | 29.57 | 75.77 | 0.41 | 83.30 | 55.83 |

**Table R3**. Results of quantitative accuracy assessment of GISA-10m for three level of cities: Level 1 (population<250,000), Level 2 (250,000 to 1,000,000), and Level 3 (>1,000,000). OA represents the overall accuracy.

| Level of cities | OA (%) | Kappa | F-Score of ISA (%) | F-Score of NISA (%) |
|---|---|---|---|---|
| Level 1 | 85.35 | 0.2205 | 91.92 | 30.41 |
| Level 2 | 87.43 | 0.2189 | 93.11 | 29.41 |
| Level 3 | 85.42 | 0.4005 | 91.86 | 47.06 |

Reference:

Characteristics and Air Quality in East Asia from 2000 to 2010, Environ. Sci. Technol., 50(17), 9142–9149, doi:10.1021/acs.est.6b02549, 2016.

Yang, Q., Huang, X. and Tang, Q.: The footprint of urban heat island effect in 302 Chinese cities: Temporal trends and associated factors, Sci. Total Environ., 655, 652–662, doi:10.1016/j.scitotenv.2018.11.171, 2019.

Line 379. How did you divide the rural and urban areas?

R: Thanks for your comment. We divided the terrestrial surface into rural and urban area using the global urban boundaries provided by Li et al., (2020).

Reference:

Li, X., Gong, P., Zhou, Y., Wang, J., Bai, Y., Chen, B., Hu, T., Xiao, Y., Xu, B., Yang, J., Liu, X., Cai, W., Huang, H., Wu, T., Wang, X., Lin, P., Li, X., Chen, J., He, C., Li, X., Yu, L., Clinton, N. and Zhu, Z.: Mapping global urban boundaries from the global artificial impervious area (GAIA) data, Environ. Res. Lett., 15(9), 94044, doi:10.1088/1748-9326/ab9be3, 2020.

Line 380. What do you mean by Global ISA?

R: Thanks for your comment. "Global ISA" refers to the global impervious surface area revealed by GISA-10m. The corresponding sentence has been revised as: " *Global impervious surface area was mainly distributed in Asia (41.43%), North America (20.59%), Europe (18.93%), followed by Africa (9.78%) and South America (7.50%).*".

Figure 14. The title of subgraph seems incorrect.

R: Corrected.

Figures 16 and 17 may be moved to the supplements.

R: Done.

Line 523. "difference" or " differences"

R: Corrected.

Line 500: "distinguish well ISA from NISA" -> "distinguish ISA from NISA effectively"

R: Corrected.

---

## Author Response (AR2)

Dear Topical Editor,

Thank you for your comments and suggestions. We have carefully revised and improved all language of manuscript and supplement, and invited a native English language editor to further improve the manuscript. The certification for the language polish has been attached below. According to your suggestion, we have reduced the length of the manuscript by 5 pages (~ 12%).

Once again, thank you very much for your time.

All the Best,
The Authors

[Figure]

Mark Ackerley
47 East Street
Leven
Beverley
HU17 5NG
Tel: +44 (0)7894 338787
mark@wordrite.co.uk
www.wordrite.co.uk

Member Chartered Institute of Editing and Proofreading (CIEP)

20 July 2022

Dear Sirs

Re: Mapping 10-m resolution global impervious surface area (GISA-10m) using multi-source geospatial data

By: Xin Huang, Jie Yang, Wenrui Wang, and Zhengrong Liu

This letter is to confirm that I have edited the above-named document.

Regards

Mark Ackerley